# Revised cytoarchitectonic mapping of the human premotor cortex identifies seven areas and refines the localisation of frontal eye fields

Sabine H. Ruland [1,4] ✉, Benjamin Sigl[2,4], Jeanette Stangier[2], Svenja Caspers[1,3], Sebastian Bludau [1], Hartmut Mohlberg[1], Peter Pieperhoff [1] & Katrin Amunts [1,2]

The premotor cortex is involved in a variety of motor and cognitive functions that often cannot be unambiguously linked to its microstructural correlates. We re-analysed the cytoarchitecture of this region in ten post mortem brains using an observer-independent mapping approach. Seven areas (6d1-6d3, 6v1-6v3, 6r1) were identified. Based on their cytoarchitectonic similarity, they were grouped into three dorsal (6d1-3) and four ventral (6v1-3, 6r1) premotor areas, supporting the concept of a functionally distinct dorsal and ventral premotor cortex. The superior frontal sulcus was identified as landmark for this separation. Comparison of the new maps with functional studies indicates that the frontal and inferior frontal eye fields are located within the premotor cortex, specifically in areas 6v1 and 6v2, not in the prefrontal cortex. Functional profiles of the areas were determined, serving as an initial basis for a more detailed characterisation of the individual areas. The new maps are publicly available to inform neuroimaging studies and aiding clinical applications such as targeting lesions or tumors and avoiding motor or cognitive impairments.

The premotor cortex (PM) is involved in motor preparation and execution[1], and more recently, in cognitive functions such as spatial perception, action understanding, imitation, cognitive manipulation, prediction and attention[2–4]. In this context, the frontal eye fields (FEFs), crucial for saccades and visuospatial attention, are of particular interest. However, their exact location and microstructural correlates are debated and associated with the dorsolateral prefrontal cortex (DLPFC) and/or the PM[5–7]. A more precise localisation of these functional activations could shed light on concepts of motor control and cognitive function, and contribute to a deeper understanding of structure-function relationships.

The broad spectrum of PM functions suggests that it is also structurally heterogeneous, comprising multiple distinct areas. However, historical and recent maps only partially capture this division, with significant variation in the number of areas (Table 1), their locations, extent, neighbouring relationships and ontology. Brodmann's map[8], still widely used as a reference in functional imaging studies, shows a single cytoarchitectonic area, the lateral part of Brodmann's area 6 (BA6). More than two decades ago, area 6 was

mapped in ten post mortem brains using observer-independent mapping[9]. However, the concept of a single area fails to account for the specific cytoarchitectonic correlates underlying distinct functional activations. Vogt and Vogt[10] subdivided the PM into four myeloarchitectonic areas. However, their map is based on only a few brains and only a schematic drawing of a 'typical' brain's surface. More recently, Fabbri-Destro and Rizzolatti[11] proposed a parcellation into six areas based on homologies with the macaque monkey. Their map is also a schematic drawing without information on the location of the borders between these areas. Using magnetic resonance imaging (MRI), Glasser et al.[12] combined the analysis of cortical structure, function and connectivity, resulting in seven areas. Fan et al.[13] subdivided the PM into four areas based on anatomical and functional connections revealed by MRI. These detailed maps show PM areas in a common 3D reference space, but they do not provide a link to the underlying microstructure, either at the individual or population-based level. Comparison between different modalities and subjects is complicated by interindividual differences regarding the sulcal pattern, microstructure and also function[14].

[1]Institute of Neuroscience and Medicine (INM-1), Research Centre Jülich, Jülich, Germany. [2]Cécile and Oskar Vogt Institute for Brain Research, Medical Faculty and University Hospital Düsseldorf, Heinrich Heine University Düsseldorf, Düsseldorf, Germany. [3]Institute for Anatomy I, Medical Faculty and University Hospital Düsseldorf, Heinrich Heine University Düsseldorf, Düsseldorf, Germany. [4]These authors contributed equally: Sabine H. Ruland, Benjamin Sigl. ✉e-mail: s.ruland@fz-juelich.de

**Table 1 | Parcellation schemes of the human PM by Brodmann[8], Vogt and Vogt[10], Fabbri-Destro and Rizzolatti[11], Glasser et al.[12] and Fan et al.[13]**

| Brodmann (1909) | Lateral part of BA6 |
| --- | --- |
| Vogt & Vogt[10] | 6aα, 6aβ, 6bα, 6bβ |
| Fabbri-Destro & Rizzolatti[11] | prePMd, PMd, FEF, PMv (F4), PMv (F5c), PMv (F5p) |
| Glasser et al.[12] | 6d, 6a, FEF, 55b, PEF, 6v, 6r |
| Fan et al.[13] | A6dl, A6cdl, A6vl, A6cvl |

*BA* Brodmann area, *PMd* dorsal premotor cortex, *PMv* ventral premotor cortex, *FEF* frontal eye field, *6d* dorsal area 6, *6a* anterior area 6, *PEF* premotor eye field, *6v* ventral area 6, *6r* rostral area 6, *A6dl* dorsolateral area 6, *A6cdl* caudal dorsolateral area 6, *A6vl* ventrolateral area 6, *A6cvl* caudal ventrolateral area 6.

For example, the shape and course of the precentral sulcus (preCS), including its junctions with the inferior and superior frontal sulci (IFS, SFS), vary considerably between brains[15,16].

In particular, the exact location of the rostral border between the agranular PM areas and (dys-)granular areas 8, 9 and 46 has not yet been mapped with modern techniques. Previous cytoarchitectonic studies have reported a high interindividual variability in this border's location[9], with no reliable macroanatomical landmark to indicate its localisation. This border is also functionally relevant, as it separates the PM from the DLPFC[17,18]. Due to the difficulty in determining the location of this border, several MRI studies limited the PM to the preCG and therefore underestimated its true extent[19,20]. This may also have affected studies investigating cognitive functions such as attention, working memory and cognitive control, which have attributed activations on the posterior middle and superior frontal gyri (MFG, SFG) to the DLPFC[21–30], although being located within the boundaries of the PM. Thus, a microstructural mapping study is needed to enable precise localisation of functional activations.

The interindividual variability in this region also poses a challenge for investigations of structure-function relationships of the FEFs. Two FEFs have been described in the human lateral frontal cortex: The FEF[5,7] and the inferior frontal eye field (iFEF[31]) or premotor eye field (PEF[6]). The FEF has been inconsistently assigned to either BA6 and/or BA8, although the tendency is increasingly towards localisation in the PM[5–7,32]. In the map by Glasser et al.[12], the FEF and PEF are defined as areas of the PM. But this map does not provide information on the interindividual variability of these areas and their rostral borders. For example, Pallud et al.[33] identified the FEFs in patients by intraoperative direct stimulation and compared their locations with Glasser's map[12]. They were localised within the FEF and PEF, but also in neighbouring areas of the DLPFC, which could indicate a more rostral localisation of the FEF in some individuals that is not captured in Glasser's map.

A division of the PM into a dorsal and a ventral part (PMd and PMv) is a well-confirmed structural and functional concept in the monkey[34]. It is assumed that the human brain has a homologous organisation. While the PMd is involved in grasping and object manipulation, PMv plays a role in reaching and action selection[35]. Various hypotheses have been formulated where the border between the PMd and PMv is localised in humans[19,20,34,36,37]. The results vary widely on the dorso-ventral axis, ranging from the level of the SFS to the IFS. This has led to activation coordinates not being uniformly assigned to the PMd or PMv. Activations of the PM located at the same dorso-ventral level of the preCG were assigned either to the PMv, or to the PMd in different studies (compare e.g. Genon et al.[37], Schubotz et al.[38], Fornia et al.[39] and Genon et al.[40]).

To gain a deeper understanding of the organisation of the PM and the relationship of functional activations to underlying microstructural correlates, we aimed (i) to map the PM in histological sections of ten post mortem brains based on cytoarchitectonic differences as captured by image analysis and statistical tests, (ii) to compute probabilistic maps (pmaps) in 3D space, and (iii) to validate the utility of these maps for studying structure-function

relationships by comparing them with results of functional imaging studies from the literature.

## Results

Seven areas, 6d1-3, 6v1-3 and 6r1 were mapped in serial sections. Cytoarchitecture was characterised by the Grey Level Index (GLI), a measure of cell packing density obtained from digitised histological images. GLI profiles running over the cortical ribbon allowed the identification of borders between cytoarchitectonic areas based on image analysis and multivariate statistical tests[41] and the quantitative description of the areas (Fig. 1A, Supplementary Fig. 1).

### Cytoarchitectonic profiling of PM areas

Areas 6d1-6d3 and 6v1-6v3 were agranular, lacking granular layer IV. Area 6r1 showed a discontinuous, very thin layer IV, i.e. was dysgranular rather than agranular (Fig. 1). The more dorsally located areas 6d1-6d3 showed a weaker layering indicated by flatter GLI profiles than the more ventral areas 6v1-3 and 6r1. Layer II was more cell-dense and the border to layer III was sharper in the dorsal than in the ventral areas 6v1-3 and 6r1. Sublayer IIIb of 6d1-6d3 showed a lower cell density and was therefore easy to distinguish from neighbouring layers. In general, the cells in areas 6v1-3 and 6r1 were more arranged in vertically orientated columns as in 6d1-6d3.

Area 6d1 was characterised by more densely packed layers IIIa, b and especially IIIc, expressed by higher GLI values compared to those to 6d2 and 6d3. The border between layers V and VI and the transition between layer VI and the white matter (WM) was rather unsharp. Area 6d2 showed less densely packed cells in the lower part of layer V and the border between layers II and III was unsharp. Layer VI showed a higher cell density, as in 6d1 and 6d3, resulting in a sharper layer VI-WM border. The highest cell density in this area (highest GLI value) was found in layer IIIc. Area 6d3 had an overall lower density, but cells of a larger size. Layer IIIc contained the largest pyramidal cells, with some of them forming clusters of three to six cells. Layer II was sharply delineable because of a low cell density in layer IIIa. Layer V showed a high density by a mixture of cells of large to small size.

The most striking characteristic of area 6v1 was its homogeneity in cell packing density across layers compared with the other ventral PM (PMv) areas. The highest GLI values were detected in layers IIIc, where also the biggest pyramidal cells were located. The border between layers II and III, as well as between III and V was less sharp. Area 6v2 showed a lower cell density in layer IIIc and therefore not such a prominent GLI peak in this layer as 6v1-6v3 and 6r1 did. In addition, the lower part of layer V was cell-sparse and easily distinguished from layer VI. While 6v1-6v3 and 6r1 showed larger pyramidal cells in layer III than in V, those of 6v2 were more similar in size, but smaller than in the other ventral areas. The highest cell density was found in layer II. The cells in 6v3 were less homogeneously distributed within the layers, and were arranged more in columns of cells. The highest cell density was found in layer IIIc, caused by a lot of medium-sized pyramidal cells. The border between layers V and VI was rather unsharp. In contrast to the other areas of the PM areas, area 6r1 showed a barely recognisable, thin and discontinuous layer IV and could thus be described as on the transition from a- to dysgranular. The loosely packed granular cells of layer IV were often interrupted by pyramidal cells of layers III and V. Sublayer IIIa was cell-dense with small pyramidal cells of different size. The transition of layer II to III was relatively seamless. The cell density in layer VI was higher than in the other PMv areas.

### Topography and relationship to macroanatomical landmarks

The PM areas occupied the preCG, the preCS, the caudal part of the SFS, as well as the caudal parts of the SFG and MFG. The extensions of areas in the rostral direction on the SFS and MFG varied between brains (Fig. 2, Supplementary Fig. 2).

Areas 6d1-3 showed a rostro-caudal as well as dorso-ventral arrangement. Area 6d1 was located on the preCG and within the preCS and SFS. It ended dorsally just before the interhemispheric fissure (IF) and ventrally mainly on the ventral bank of the SFS. Its caudal border was located on the

**Fig. 1 | Observer-independent border detection and profiling of PM areas. A** Region of interest of a cell-body stained section with a border (orange line) between areas 4a and 6d1 (top left). The black rectangles mark the positions of the zoom-ins below, showing the cytoarchitecture of 4a (bottom row, left) and 6d1 (bottom row, right), as well as the cell packing density over the cortical layers II–VI quantified by a grey level index (GLI) profile (black and green line). Roman numerals (I–VI) indicate cortical layers. Area 4a mainly differs from 6d1 by Betz cells in layer V, an unsharp layer VI-WM border and a broader layer VI, which is subdivided into two sublayers. For the observer-independent border detection, GLI profiles are extracted along transverses (1–162) along the cortical ribbon (top right). Averaged profiles of two adjacent blocks of profiles, including 12–24 profiles per block, are calculated to increase the signal-to-noise ratio. The Mahalanobis distances (MD) as a measurement for cytoarchitectonic differences is computed by calculating the distances between all pairs of neighbouring cortical blocks using a sliding window approach. These values are plotted as a function of the profile position. The upper graph shows this function using a block size of 23 profiles. A significant MD maximum ($p < 0.001$) is found at profile position 76. The graph below shows additional significant MD maxima at the profile position 76 using blocks of 16–24 profiles, indicating a border between neighbouring areas at this position. Additional borders between the PM areas and neighbouring areas are shown in Supplementary Fig. 1. **B** Cytoarchitecture of areas 6d2, 6d3, 6v1, 6v2, 6v3 and 6r1 and corresponding GLI profiles.

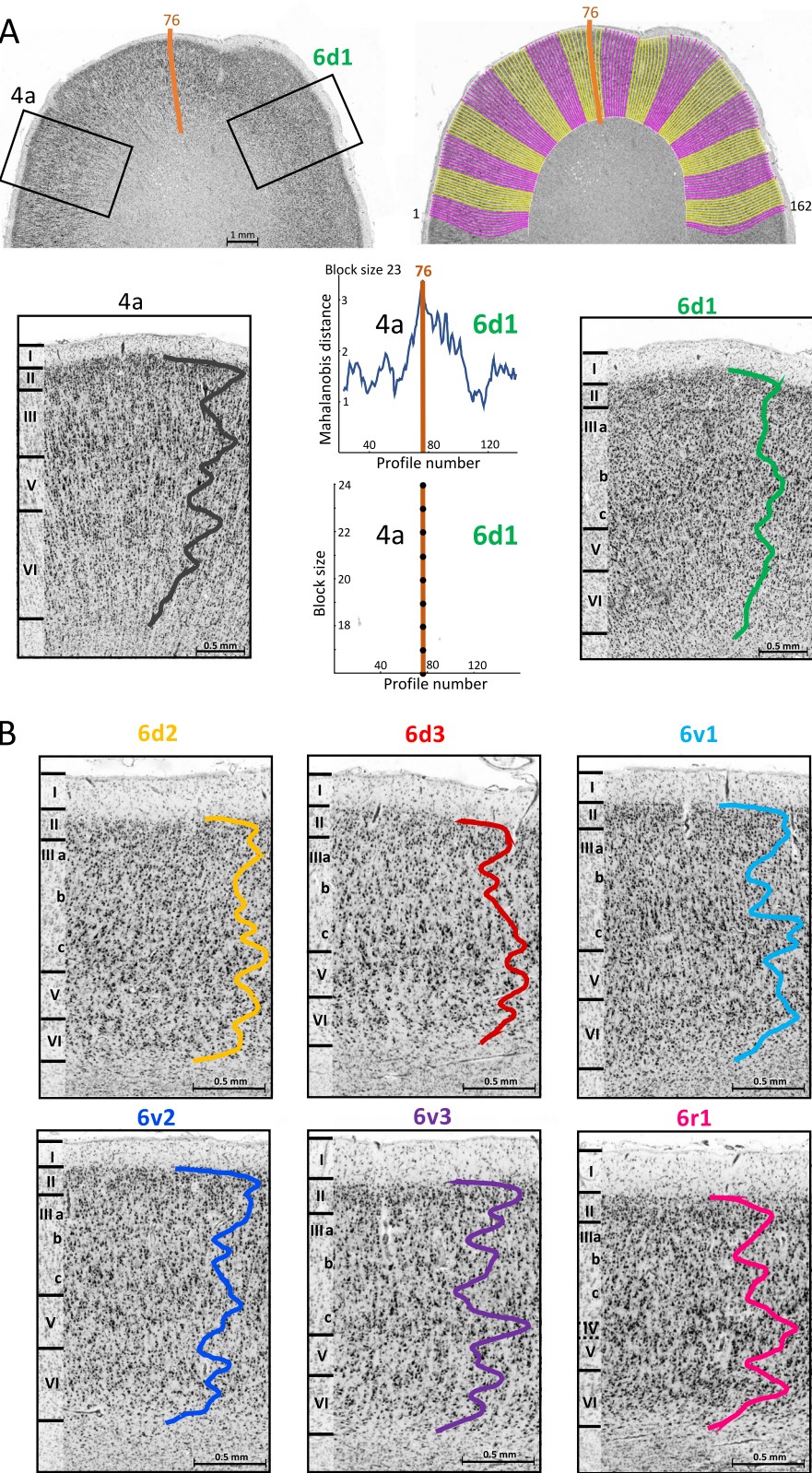

preCG just before the crown of the central sulcus (CS). The rostral border was found at the transition between the preCG and the SFG. In about 75% of the cases, a vertically oriented segment of the superior preCS separating the preCG from the SFS served as the approximate landmark of the rostral border of 6d1–6d2 (e.g. Fig. 2, BC01, BC05 L, BC09, BC10 R, BC11, B20 L). This was the case when the superior preCS was a continuous sulcus and not separated into several segments[42]. Area 6d2, located rostral to 6d1, was located in the caudal part of the SFG and extended from dorsally just before the IF to ventrally, mainly on the dorsal bank of the SFS, where it bordered area 6d3. Its caudo-rostral extension was variable between the brains. Its rostral border was not marked by any macroanatomical landmark. E.g. 6d2 in BC11 R showed a broad extension on the caudo-rostral axis, but in

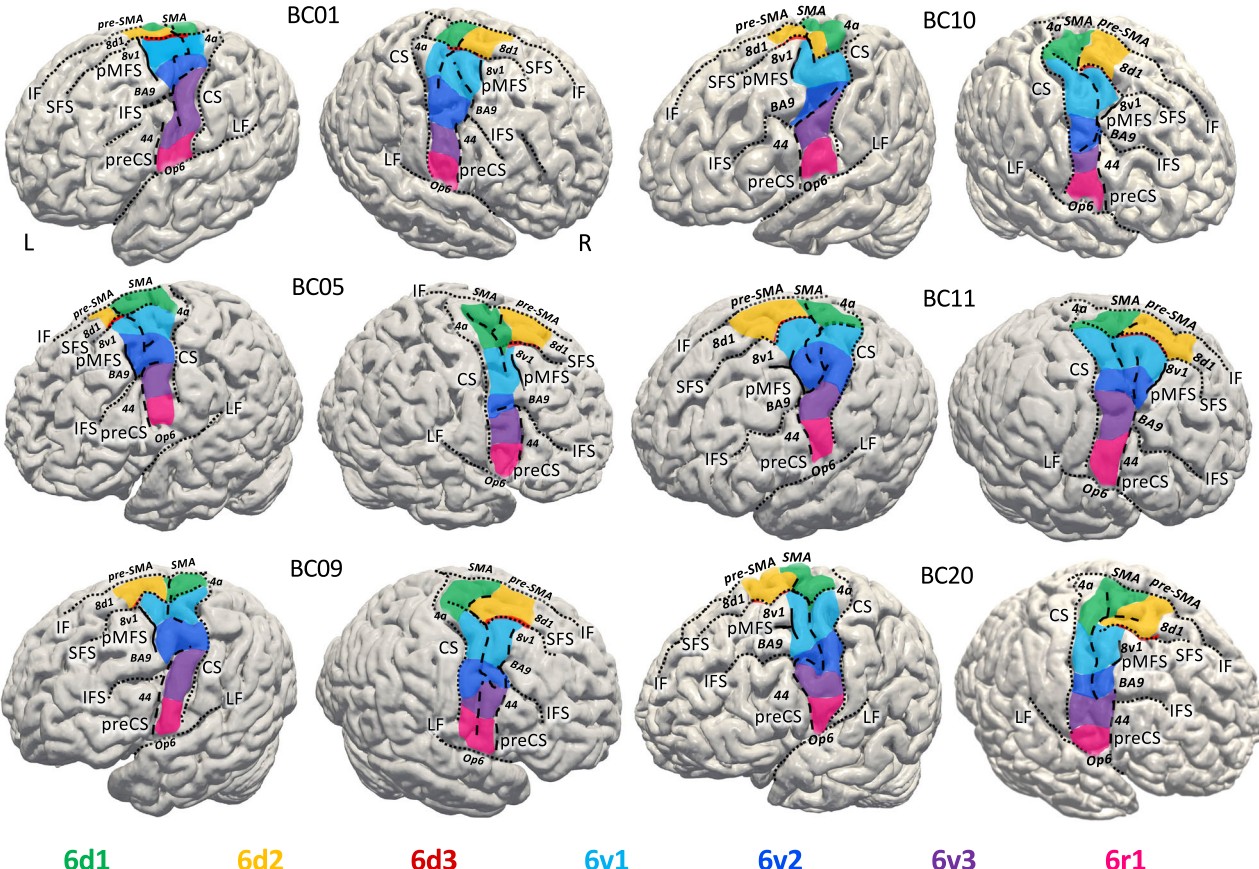

**6d1**  **6d2**  **6d3**  **6v1**  **6v2**  **6v3**  **6r1**

**Fig. 2 | Individual maps of six post mortem brains. A** 3D reconstruction and maps of PM areas on the pial surfaces of six brains illustrating the interindividual variability. Area 6d3 is almost not visible on the surface because it is located within the SFS. Sulci are marked by black (dotted) lines. BC brain code, CS central sulcus, IF interhemispheric fissure, IFS inferior frontal sulcus, LF lateral fissure, preCS precentral sulcus, pMFG posterior segment of the middle frontal sulcus, SFS superior frontal sulcus. The preCS is interrupted and subdivided into several parts. Neighbouring areas are indicated in bold italics. Caudally, the PM areas bordered the

primary motor cortex, area 4a[114] (Fig. 1A). Medially to areas 6d1 and 6d2, areas SMA and pre-SMA[117] were located. Rostro-dorsally, areas 6d2, 6d3 and 6v1 shared borders with areas 8d1, 8d2 (Supplementary Fig. 1), and 8v1[111] of the posterior dorsolateral prefrontal cortex. More ventrally, 6v1, 6v2 and 6v3 bordered BA9. At the level of the inferior frontal sulcus, 6v2 and 6v3 were found in the neighbourhood to area ifj2[116]. Area 44 of Broca's region[115] was located rostrally to areas 6v3 and 6r1. Most ventrally, area 6r1 bordered the frontal operculum, area Op6[118]. L left hemisphere, R right hemisphere. Mappings of remaining brains are shown in Supplementary Fig. 2.

BC05 L a rather narrow one. Area 6d3 was only found in the depth of the SFS, ventral to 6d2 and rostral to 6d1. Its caudal end was found in the junction of the SFS and preCS, its rostral one approximately on the same level of the dorso-ventral axis as the rostral border of 6d2. The caudal part of the SFS which included 6d3 was continuous and not interrupted in all studied brains.

Areas 6v1, 6v2, 6v3 and 6r1 were arranged in a dorso-ventral sequence from the SFS to the LF. These areas ended caudally on the transition of the rostral part of the CS on the preCG. The most dorsally located area 6v1 abutted 6d2 and 6d3 mainly on the ventral wall of the SFS (or its extension caudally to the preCS). Area 6v2 was located ventrally to 6v1. The border between these areas was not marked by any sulcus and its location was variable on the dorso-ventral axis in the different hemispheres. E.g. in BC05 R and BC09 R, this border was found on a more ventral level than in BC05 R or BC09 R. Areas 6v1 and 6v2 covered the preCG and extended rostrally on the caudal part of the MFG in 19 of the 20 investigated hemispheres. In these hemispheres, the border of 6v1 and 6v2 to the neighbouring areas was indicated by a mainly dorso-ventrally running posterior segment of the medial frontal sulcus (pMFS), which directly followed the preCS rostrally (Fig. 2 BC01, BC05; BC09 L, BC10, BC11, BC20). This segment was sometimes connected to neighbouring sulci, e.g. the IFS, close to its junction with the preCS (e.g. BC05 L, BC11 L) or to the SFS (e.g. BC11 R, BC09 L). The only exception where the PM areas did not reach on the MFG was the right hemisphere of brain BC09. This hemisphere showed a double parallel

type of the preCS[15]. In this case, the rostral border of the PM was marked by the rostrally located preCS. In general, the location of pMFS which marked the rostral border of areas 6v1 and 6v2 to neighbouring cortices, showed a high variability in its rostro-caudal localisation between the hemispheres resulting also in highly variable rostral extensions of areas 6v1 and 6v2 in the different brains. For example, these areas showed a more rostrally located border in brain BC11 than in brain BC05. The border between 6v2 and 6v3 was found in most cases within a caudally directed horizontal extension of the dorsal component of the inferior preCS[42] (Fig. 2 BC01 L, BC05, BC09 L, BC10, BC11, BC20 L; Supplementary Fig. 2 BC04, BC07, BC08 L, BC21). The border was therefore located approximately on the level of the inferior frontal sulcus (IFS). The horizontal extension of the dorsal component of the inferior preCS on the preCG showed a variable pattern. This sulcus was running either dorsally (e.g. BC01 L, BC10 L, BC11 L), ventrally (BC10 R, BC20 L) or did not show any remarkable changes in the dorsal or ventral direction (BC05; BC09 L). This also influenced the course of the 6v2-6v3 border on the surface accordingly. In addition, in hemispheres which showed a horizontal extension that was very deep, large portions of 6v2 and/ or 6v3 were located within the sulcus and less on the preCG surface (e.g. Fig. 2 BC10 L, BC05 R, Supplementary Fig. 2 BC21 R, BC07 R). The rostral border of 6v3 was detected within the ventral component of the inferior preCS, which was in all brains a continuous sulcus without interruptions. 6v3 was never found on neighbouring gyri. 6r1 was the most ventrally located area of the PM. The border between 6r1 and 6v3 was found on the

preCG approximately in the middle between the level of the IFS and the LS. Area 6r1 was only found on the preCG with its rostral border in the depth of the ventral component of the inferior preCS. The ventral border was identified dorsally to the LF, bordering the frontal operculum.

### 3D probabilistic maps in the Julich-Brain Atlas

The maps of the individual brains were warped to the Colin27 and ICBM152 non-linear asymmetric 2009c template to compute pmaps which quantify the interindividual variability in extent and localisation of the areas (Fig. 3A). Probability maps of areas 6v1 and 6v2 showed 'more blue voxels' than the maps of the other areas, indicating a higher variability. The maps are part of the Julich-Brain Atlas and freely available on EBRAINS.

Table 2 shows the centres of gravity of the probability maps in MNI space (Colin27) and the volumes of the areas. The volumes did not differ between the left and right hemispheres ($p > 0.05$), and no gender differences were found ($p > 0.05$).

To provide a simplified version of the probability maps, where each position of the reference brain is linked to a single area, we then computed maximum probability maps (MPMs) of the areas (Fig. 3B).

### Cytoarchitectonic similarities estimated by multidimensional scaling and cluster analysis

A multidimensional scaling (MDS) and a hierarchical cluster analysis were used to compare the cytoarchitecture of the PM areas among themselves and with neighbouring areas. The shorter the distance between areas, the more similar their cytoarchitecture (Fig. 4). Areas 6d1-6d3 showed more similarities among each other than each of them to 6v1, 6v2, 6v3 and 6r1, which also formed a distinct cluster. Based on these results, areas 6d1, 6d2 and 6d3 were grouped as the dorsal PM (PMd). The same applied to 6v1, 6v2, 6v3 and 6r1, which formed the PMv. In addition, the PMd areas shared more similarities with the primary motor cortex than the PMv areas. The PMv areas were more similar to areas 44 and 45 of the inferior frontal gyrus.

### Localisation of borders of PM areas to neighbouring cortices in stereotaxic space

The PM appeared as a wedge-shaped stripe, which was wide at its dorsal part and increasingly tapered towards its ventral end. The localisation of the rostral border of areas 6v1 and 6v2 on the MFG extended to a maximum level of $y = 13$ in the left hemisphere and $y = 17$ in the right hemisphere (Fig. 3).

In accordance with the results of the MDS (Fig. 4), the border between the PMd and PMv areas was formed by the border between 6d1/6v1 in the caudal and 6d3/6v1 in the rostral PM. This border coincided with the caudal extension of the SFS (Figs. 2, 3 and 5B).

### Spatial comparison of cytoarchitectonic maps with results from functional studies

To explore the functional role of the new areas, we compared the results of functional studies from the literature with the new maps. Figure 6A shows left hemispheric coordinates reported for the FEF in 30 studies[43–71], superimposed to the cytoarchitectonic maps. Figure 6B showed left hemispheric coordinates reported for the iFEF in 9 studies[6,31,65,67–72]. FEF coordinates showed a spatial variability on the $x$-axis in the range of 14 mm, on the $y$-axis of 25 mm and on the $z$-axis of 17 mm (Supplementary Table 1). Seven of the 30 coordinates could not be localised within any pmap of Julich-Brain. The remaining twenty-three coordinates could be localised within pmaps of PM areas. Eleven coordinates were located within 6d1, eight within 6v1, seven within 6d3 and one within 6v2. The iFEF coordinates (Fig. 6B, Supplementary Table 1) were located ventrally to the FEF coordinates. One coordinate could not be assigned to any pmap, five coordinates were localised within 6v2, three within 6v3 and two within areas 4a, 4p and 3b.

In addition to comparing coordinates of peak activations, areas FEF and PEF of the map by Glasser et al.[12] were correlated with Julich-Brain areas. The comparison revealed the largest correlation of the FEF area with 6v1 and of the PEF area with 6v2 (Table 3).

When plotting the left hemispheric activation coordinates from studies on the execution of motion of the foot/leg[73,74], arm resp. reaching activity[75–79], hand resp. grasping activity[80–83] and mouth[20,84] on the maps of the PM (Fig. 6C), activations of the motion of leg/foot as well as the arm were found in areas of the PMd, mainly 6d1 and 6d2, and activations of mouth and hand motions in area 6v3.

Finally, we plotted coordinates of peak activations reported in Schubotz et al. 2001 and 2003[2,38,85], using a serial prediction task with visual and auditory stimuli (Fig. 6D). The peak activations found in the spatial prediction tasks using visual and auditory stimuli were located in 6d1, the activations of object-related prediction mainly in 6v2 and 6v3 and those of temporal prediction in areas of the frontal Operculum (Op6, Op8) and area 44. In general, attending to visual stimuli were located in area 6v3 and auditory ones in Op6.

Figure 6E summarises the results of the spatial comparisons with functional activation coordinates described above and presents initial functional profiles of the PM areas.

## Discussion

This study presents maps of seven cytoarchitectonic areas in the human PM, providing information about spatial localisation and interindividual variability. In accordance with a functional subdivision into PMd and PMv[11,34], we identified two groups of areas based on cytoarchitectonic similarities. Thus, cytoarchitecture supports the hypothesis of a dorso-ventral subdivision of the PM, but also shows that each subdivision is composed of multiple areas. The SFS serves as an anatomical landmark of this subdivision. This aligns with PMd-PMv borders from comparative studies [3], though functional and connectivity studies show varying results at more ventral levels[19,20,36,40].

Historical and more recent maps of the PM show both similarities but also notable differences to the present map, particularly in the number, arrangement and extension of areas[8,10–13]. For example, the map by Glasser et al.[12] shows an equal number of PM areas like our parcellation, but with some differences in their location and arrangement. The most dorsally located areas 6d and 6a show a similar rostro-caudal arrangement like 6d1 and 6d2 in our study, but the borders of 6d and 6a to the medially located areas of the supplementary motor cortex (6ma and 6mp) are located further on the lateral surface than those of 6d1 and 6d2. Therefore, area 6d1 shows a great correlation with Glasser's area 6mp and to a lesser extent with 6d. Area 6d2 largely correlates with 6ma. Area 6d3 largely corresponds to 6a. Areas 6v1-3 and 6r1 show a dorso-ventrally arrangement, whereas areas FEF, 55b, PEF, 6v and 6r are also located rostro-caudally to each other. Area 6v is located caudally to PEF and 6r. The FEF correlates mainly with 6v1. Both 55b and PEF seem to lie within 6v2. Area 6v has a great correlation with 6v3 and 6r with 6r1. The map by Fan et al.[13] includes a smaller number of areas. The location of A6dl on the SFG seems to correspond mainly to 6d2 and 6d3. A6vl, located on the MFG, ventrally to A6dl, may cover rostral parts of our areas 6v1 and 6v2. A6cdl is situated caudally to A6dl and A6vl on the preCG and matches the position of 6d1 and the caudal parts of 6v1 and 6v2. A6vcl covers the preCG ventrally to the level of the IFS, containing areas 6v3 and 6r1 of our map. Discrepancies between the different maps may be due to brain variability or methodological differences. Historical maps were based on pure visual inspection of only one or a few hemispheres. This can make it difficult to distinguish cytoarchitecturally similar areas, like in the PM. In our study, the use of statistical criteria in a sample of ten brains led to a reproducible identification of areas, even when their locations and volumes vary between the brains. The high interindividual variability in this region[9], is now quantified by pmaps. While MR studies lack the resolution of histology, they allow for larger sample sizes, which are analysed at the group level. However, a key challenge in group-level analysis is the spatial interindividual variability of areas[14,86]. Brain areas with less variability are more likely to reach statistical significance, whereas areas with higher variability may not, which can lead to false-positive or negative results. Additionally, the application of different pipelines can produce varying results depending on spatial smoothness, the used software package and thresholding[87]. These

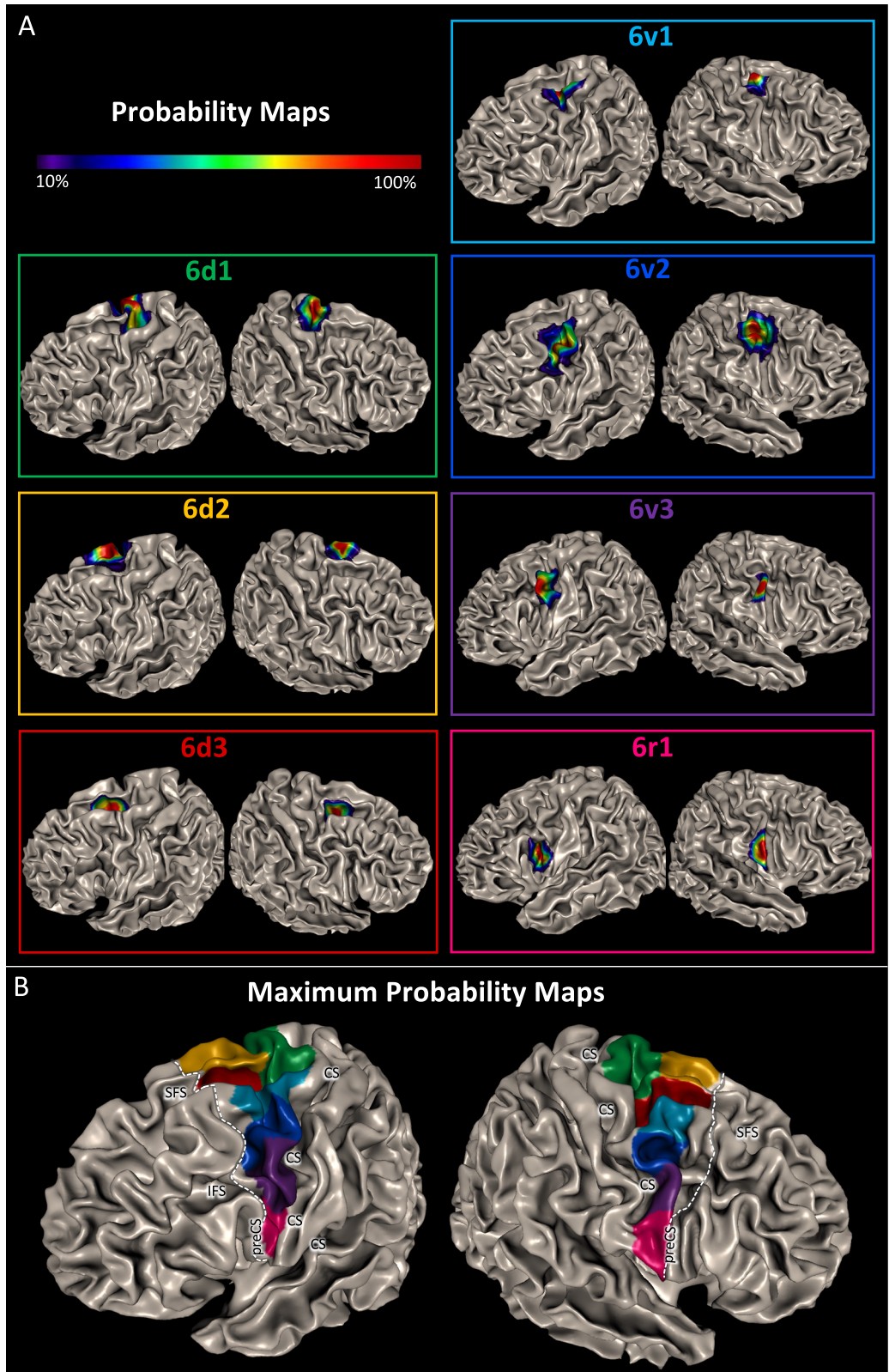

**Fig. 3 | 3D probabilistic maps and maximum probabilistic maps.**
**A** Cytoarchitectonic probability maps (pmaps) of the PM areas based on the mapping of ten post-mortem brains projected on the smooth white matter surface of the stereotaxic MNI Colin27 template brain. The colour bar indicates the probability with which a certain area was present in a given voxel. **B** Maximum probability map (MPM) of PM areas in the MNI Colin27 template (smooth white matter mode). The white dotted lines mark the maximum extension of the PM border in the rostral direction, as revealed by the pmaps of the PM areas. Maps of all PM areas are freely available in different standard template spaces on EBRAINS. CS central sulcus, IFS inferior frontal sulcus, preCS precentral sulcus, SFS superior frontal sulcus.

methodological limitations can explain the differing parcellations of the PM between MR studies[12,13] and between MR and histological approaches[8,10,11].

Our new cytoarchitectonic maps more precisely characterise the rostral border of PM areas relative to areas of the DLPFC. The pmaps provide valuable information about the maximum extension of the areas and the probability the areas can be found at a specific point in the reference brain. While these maps are based on the examination of ten brains - a smaller cohort with lower statistical power compared to typical neuroimaging studies - they are derived from high-resolution histological data. This high spatial precision enables the definition of cortical borders with far greater anatomical accuracy than is currently possible with in vivo imaging techniques. As a result, these maps offer unique insights into individual variability and extend the existing knowledge of structural organisation in this region beyond what has previously been available. When comparing the maps with previous studies, we found that the rostral extension of the PM was often underestimated. Geyer[9] limited area 6 to $y = 0$, while our study shows it extending to $y = 15$, and up to $y = 17$ in some brains. Some MR studies limited the regions-of-interest for studying PM connectivity to the preCG[19,20]. As the rostral extent of the PM was not clearly defined, this choice was made to ensure that adjacent prefrontal areas were excluded from the sample, but at the cost of excluding parts of PM areas located on the SFG and MFG. Similarly, some functional studies investigating working memory, cognitive control and attention also underestimated the rostral extension of the PM. Activations were assigned to the DLPFC, but they were actually located within the maps of areas 6v1-3[21–30]. Our analysis has also identified the pMFS as a reliable landmark for the rostral borders of areas 6v1 and 6v2 on the MFG. This result supports an earlier finding by Amiez et al.[88], who suggested, based on comparative analyses, that a posterior segment of the MFS (their posteromedial frontal sulcus) corresponds to the border between the premotor and prefrontal cortex. The new maps and coordinates, along with the insight that the rostral PM border on the MFG is associated with a sulcus, will enhance the localisation of the rostral PM border in imaging studies and allow for more anatomically precise assignments of activations to microanatomical areas.

The underestimation of the rostral extent of the PM also impacted the identification of the microanatomical substrate of the FEF. The FEF has been inconsistently located within the PM or (partly) in BA8[5,6,32]. Our study, however, found that peak activation coordinates for the FEF were all located within maps of PM areas and not in the more rostrally located areas of the DLPFC. This finding aligns with the map by Glasser et al.[12], who also placed the FEF within the PM. We observed the greatest correlation between the FEF area in the map by Glasser et al.[12] and area 6v1 mapped in our study. Also, Tehovnik et al.[89] and Preuss et al.[90] suggested the FEF to be located within the agranular cortex of the PM, more precisely within the preCS, caudal to the MFG. This macroanatomical-functional relationship for the FEF is also supported by Amiez et al.[43], demonstrating that FEF activations were located within the superior preCS, in its branch ventrally to the SFS. Similarly, Borra and Luppino[91] located the FEF within BA6, reporting a similar association with the preCS/SFS. These descriptions correspond closely with the location of 6v1 mapped in our study. When comparing the peak activation coordinates for the FEF with our maps, we did not find a clear assignment to one of the PM areas. In addition to 6v1, coordinates were also found in neighbouring 6d1 and 6d3. It is important to note that the spatial variation in the coordinates across different functional studies was substantial. This variability could stem from differences in the experimental protocols, as well as methodological factors, including spatial normalisation and alignment of individual brains to reference spaces. The interindividual variability in sulcal pattern, which is particularly high for the region of the junction of the preCS and the SFS[92,93], remains a challenge for alignment tools[94]. Amiez and Petrides[57] demonstrated in an fMRI study on saccadic eye movement that only a subject-by-subject analysis—not the group-level analysis, revealed the anatomo-functional relationship between activations

**Table 2 | Coordinates (x, y, z) of the centre of gravity of pmaps in MNI space (Colin27) and areal volumes in mm³ (mean value and standard deviation, after shrinkage correction) of 6d1, 6d2, 6d3, 6v1, 6v2, 6v3 and 6r1 in the left (L) and right (R) hemisphere**

| Area | Centres of gravity in MNI space | | | |
| | x | y | z | Volume (mm³) |
| --- | --- | --- | --- | --- |
| 6d1 L | −20 | −14 | 67 | 3996 ± 1299 |
| 6d1 R | 20 | −16 | 68 | 4048 ± 1552 |
| 6d2 L | −16 | 1 | 66 | 2909 ± 1035 |
| 6d2 R | 16 | 4 | 67 | 3112 ± 884 |
| 6d3 L | −23 | 2 | 53 | 2046 ± 900 |
| 6d3 R | 25 | −1 | 54 | 1903 ± 830 |
| 6v1 L | −36 | −2 | 56 | 4100 ± 1482 |
| 6v1 R | 38 | −4 | 57 | 4514 ± 1169 |
| 6v2 L | −43 | 3 | 42 | 2101 ± 947 |
| 6v2 R | 46 | 4 | 40 | 2164 ± 872 |
| 6v3 L | −51 | 5 | 34 | 1664 ± 570 |
| 6v3 R | 58 | 5 | 33 | 1755 ± 908 |
| 6r1 L | −54 | 8 | 18 | 1673 ± 647 |
| 6r1 R | 60 | 7 | 17 | 1752 ± 879 |

Volumes of the individual brains (n = 10 for each of the areas) are shown in Supplementary Table 3.

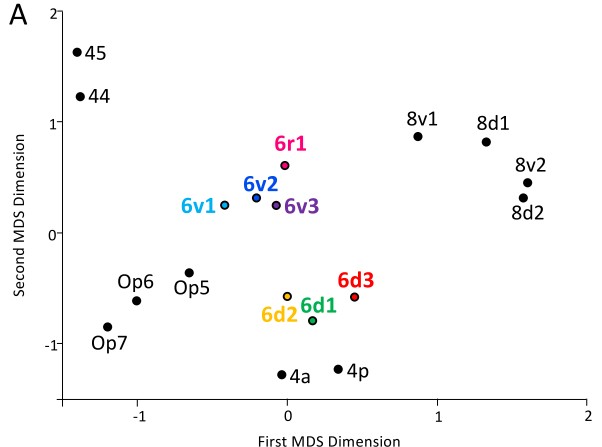

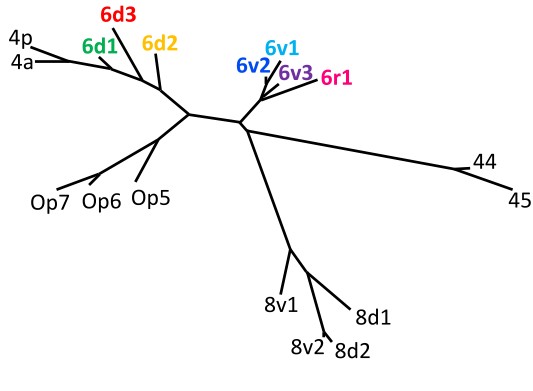

**Fig. 4 | Cytoarchitectonic similarity of PM areas and neighboring areas.** Multidimensional scaling (MDS, **A**) and cluster analysis (**B**) of the PM areas as well as their neighbouring areas of Broca's region (44, 45), the primary motor cortex (4a, 4p), the posterior dorsolateral prefrontal cortex (8v1, 8v2, 8d1, 8d2) and the frontal operculum (Op5, Op6, Op7). N = 10 brains; stress value in MDS: 0.047.

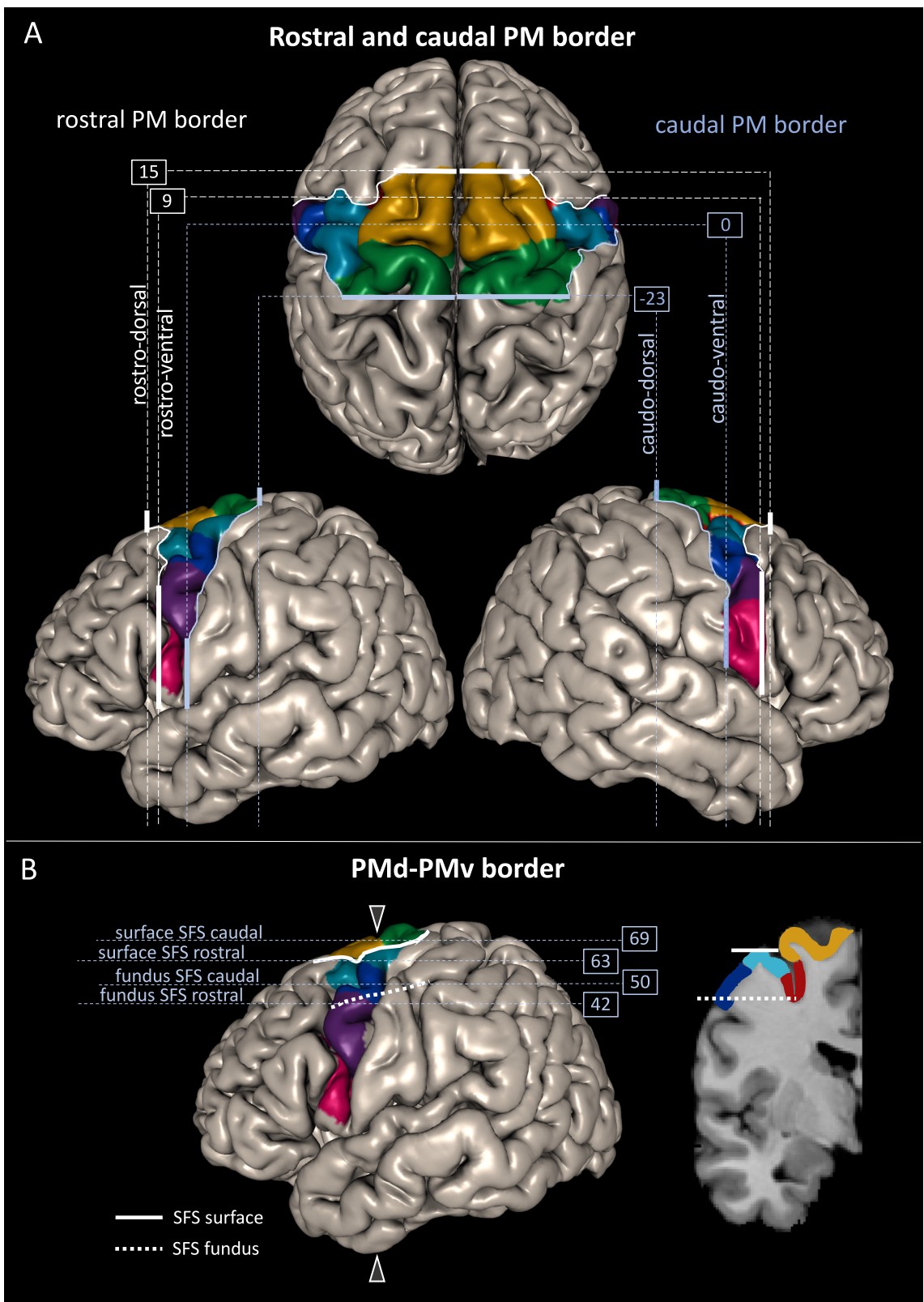

and the preCS/SFS. In summary, there is strong evidence that 6v1 serves as the structural correlate of the functionally defined FEF.

Some FEF studies reported a second activation located ventrally to the FEF[6,31,65,67–72], the iFEF or PEF. The functional dissociation and specifications of the FEF and iFEF remain poorly understood. Some authors have suggested that the iFEF may be involved in eye blinks[6]. The iFEF coordinates

were predominately found in 6v2. Amiez and Petrides[6] described the iFEF as being located within the dorsal branch of the inferior part of the preCS. Derrfuss et al.[31] reported that this activation also consistently extended into parts of the preCG, a pattern that closely corresponds to the location and extension of 6v2. Moreover, comparison with the PEF area in the map by Glasser et al.[12] points to 6v2 as the microstructural correlate of the iFEF.

**Fig. 5 | Coordinates of PM borders in MNI space. A** The rostral (white line) and caudal (light blue line) borders of the PM are quantified by y levels (dashed lines) based on the MPMs in MNI Colin27 space. The rostral border on its dorsal end (rostro-dorsal border, on the SFS) is located on the SFG at $y = 15$ mm, the rostro-ventral border is marked by the inferior preCS and was located at $y = 9$. In between, on the MFG, the border retreated a few mm more caudally. The caudo-dorsal border is located on the preCG at $y = -23$ and the caudo-ventral border, marked by the CS, at $y = 0$. In between, the border ran in a slightly rostrally orientated curve. **B** The border between the PMd and PMv is topographically marked by the SFS and located on its ventral bank as shown in coronal section (position at $y = -6$, marked by arrows

on the brain surface) on the right-hand side. The course of the SFS goes towards ventral and its fundus is located ventrally to the entrance on the surface. Taking into account this special topography of the SFS, it is necessary to define z levels for a dorsal border marked by the entrance of the SFS on the surface as well as a ventral border in the depth on the level of the fundus of the SFS. Dorsally, on the brain surface, the border is marked by the entrance of the SFS (white line). At the rostral part the border is located at $z = 63$ and at the caudal part at $z = 69$. Ventrally, in the depth, the border is marked by the fundus of the SFS (white dashed line), rostrally located at $z = 42$ and caudally at $z = 50$.

However, future studies are needed to provide a comprehensive demonstration of these structure-functional relationships using the new maps as anatomical reference. These maps will also facilitate functional studies aimed to distinguishing the roles of the FEF and iFEF in spatial attention and oculomotor control.

The seven new areas are clustered into two groups based on cytoarchitectonic similarities: three dorsal and four ventral areas. This subdivision aligns with the functional subdivision[95,96] into PMd and PMv. Comparative, functional and connectivity studies have proposed different locations for the PMd-PMv border along the dorso-ventral axis between the SFS and IFS[19,20,34,36,37]. These differing perspectives have affected how functional activations were assigned to PMd or PMv. Activations at the same dorso-ventral level on the preCG have been inconsistently attributed to either the PMd or PMv across different studies (compare e.g. Genon et al.[37] and Schubotz et al.[38], Fornia et al.[39] and Genon et al.[40]). The present cytoarchitectonic study, however, provides a clear anatomical landmark, the SFS, for the PMd-PMv border. That the SFS acts as a landmark for functional subdivisions within the PM is also supported by Mackay et al.[97], who identified two visual field maps (superior precentral sulcus 1 and 2, sPCS1/2) which were separated by the SFS. The new PM maps and coordinates for the microstructural PMd-PMv border will facilitate future studies to understand the relationship of microstructural, macrostructural and functional subdivisions.

The localisation of activations related to body motion within the newly mapped areas provides further insight into the functional specialisation of these areas. Comparing activation coordinates with cytoarchitectonic areas in 3D space revealed that leg/foot movement and reaching activate the PMd, specifically 6d1 and 6d2, while mouth movements and grasping are associated with the PMv, particularly 6v3. These results align with monkey studies, which show overlapping representations of the hindlimb and arm in the PMd, and of the forelimb and face representation within the PMv[98]. Human functional studies have also provided evidence of a similar organisation within the human PM for the observation and execution of motion of the different body parts[2,74,99–102]. Rizzolatti et al.[34] proposed a somatotopic organisation, with arm and leg representations in the PMd, and the digits, face and mouth in the PMv. This organisation is consistent with the role of the PMs in reaching and the PMv in grasping[103,104]. Our study supports this concept, suggesting that this organisational scheme is evolutionarily conserved across monkeys and humans.

In addition to the spatially distributed representations of body parts, different aspects of prediction involve distinct PM areas. Spatial prediction activations were found in 6d1, while object-related prediction were located mainly in 6v2 and 6v3. Temporal prediction peak activations were located medially, but very close to 6r1, in area 44 and the frontal operculum, with the entire activation cluster extending into the map of 6r1. Schubotz et al.[2,38,85] argued, in accordance with the premotor theory of attention[105,106], that prediction is represented within the PM in a body-referenced manner: spatial prediction activates PM areas which are also activated for saccades and reaching, object-related prediction with grasping, and temporal prediction with vocal production. Our study found similar overlaps, with spatial prediction in 6d1 and object-related prediction in 6v3. Area 6r1's involvement in temporal prediction suggests a link to language functions. Cytoarchitectonically, 6r1 is distinct from the other mapped PM areas, being

slightly dysgranular rather than agranular. This places it at the transition between the agranular PM and the dysgranular area 44 of Broca's region[107]. Functional imaging studies have shown activations in the PMv and the frontal operculum during syntactic processing, which may be related to 6r1, suggesting it is functionally more closely related to area 44 than to the PM itself[108,109]. The PMv's potential role in language is further supported by a theory, suggesting its evolution from motor programs used in primate social signalling towards voiced communication[110]. Thus, it could be speculated that 6r1 represents an evolutionary precursor to language.

Glasser et al.[12] identified an area within the PM involved in language tasks, referred to as 55b, which is located between the FEF and PEF. The present study showed that Glasser's FEF and PEF correlate with 6v1 and 6v2, respectively. Glasser's area 55b might correspond to a dorsal portion of 6v2. Notably, our study did not reveal further subdivision based on cytoarchitectonic criteria.

In conclusion, the present work: (i) provides a detailed parcellation of the PM, introducing seven new areas. (ii) Highlights hierarchical principles within the PM, offering a solid neurobiological basis for the functional subdivision into PMd and PMv. (iii) The new maps are provided in a format compatible with in vivo imaging and enable further exploration of structure-function relationships. (iv) The application of these maps has resolved a long-lasting 'enigma' in neuroscience[91]: the localisation of the human FEF. Both the FEF and iFEF are located within the limits of the agranular PM, specifically in 6v1 and 6v2. (v) Functional profiles, derived from the comparison of coordinates from functional imaging studies and the new maps, provide an initial insight into the functional specialisation of the newly defined areas, forming the basis for further functional studies. (vi) The maps also provide a valuable resource for clinical application, such as localising epilepsy foci or brain tumours, and can help avoid complications such as apraxia or cognitive impairments.

## Methods
### Histological processing
Post mortem brains for cytoarchitectonic analysis were obtained through the body donor program of the Department of Anatomy I, at the University of Düsseldorf, Germany. Brains were collected and prepared in accordance with the rules of the local ethics committee (study numbers 2023–2632). All ethical regulations relevant to human research participants were followed. The donors (Table 4) had no clinical history of neurological or psychiatric diseases. The post mortem delay was between 12 and 24 h.

Histological processing has been described in detail by Amunts et al.[111]. In short, brains were fixed in formalin or Bodian fixative, embedded in paraffin and cut into 20 μm-thick coronal sections. Every 15th section was mounted on a gelatine-covered glass slide and stained for cell bodies using a modified silver staining[112]. One of the brains (BC20) has been the so-called BigBrain with a total series of 7404 sections, that have been 3D-reconstructed with 20 μm spatial resolution isotropic[113].

### Identification of region of interest (ROI)
The region of interest (ROIs), the lateral PM was identified in images of histological sections. Sulci and gyri (CS, preCS, IFS, SFS etc.) of the surface and sections of the individual brains have been identified according to Ono et al.[15]. The lateral PM mapped in this study largely corresponds to the lateral

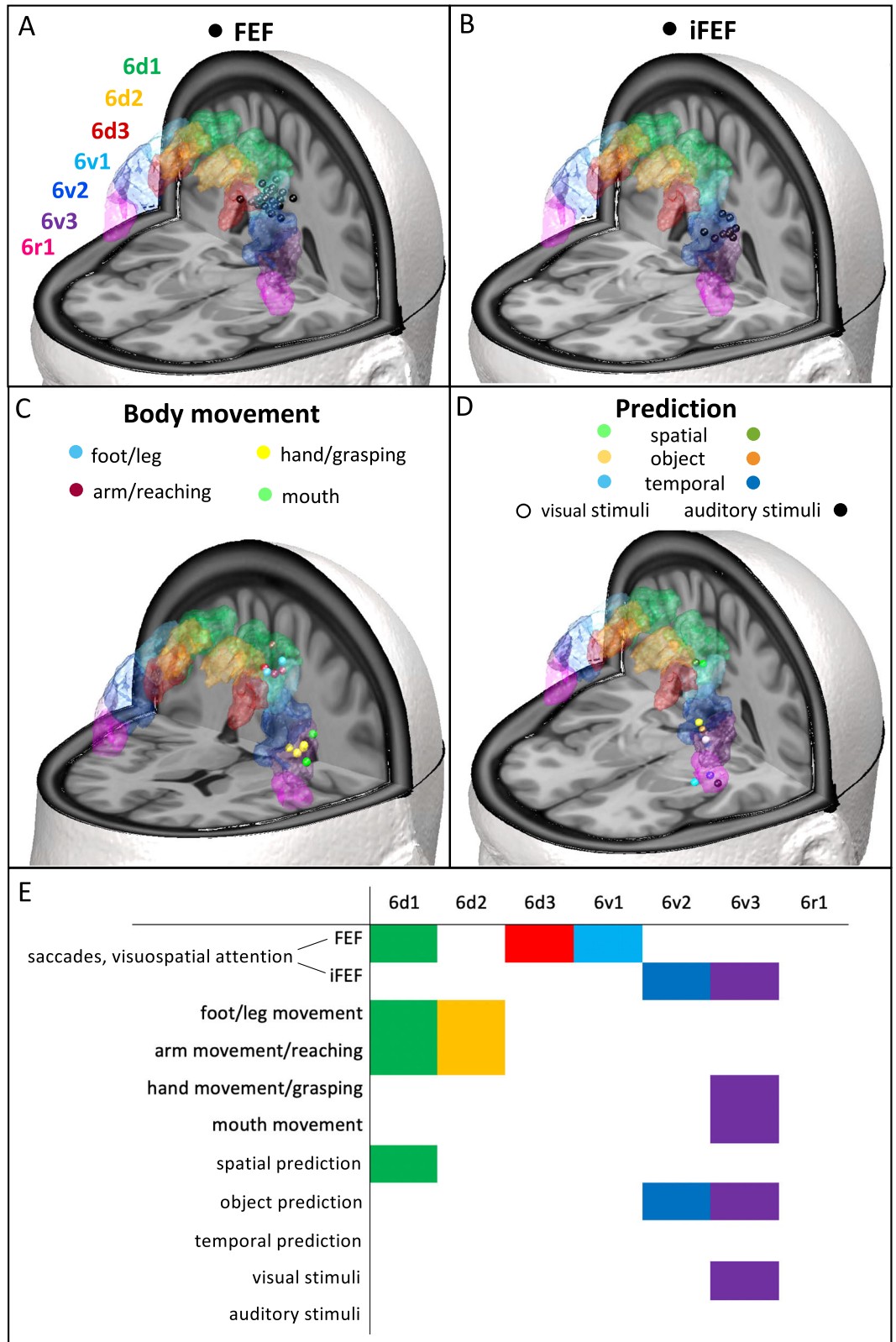

**Fig. 6 | Spatial comparison of the PM map and left hemispheric coordinates reported in functional studies on the FEF, iFEF, movement of body parts and prediction. A** Coordinates of the FEF reported in 30 studies[43–71] are plotted on the MPM of the PM areas in ICBM152 space. **B** Coordinates of the iFEF reported in 9 studies[6,31,65,67–72]. **C** Coordinates reported in studies investigating the execution of motion of the foot/leg[73,74], the arm resp. reaching activity[75–79], hand resp. grasping activity[80–83] and the mouth[20,84]. **D** Activation peaks in experiments[2,38,85] on spatial, object-related and temporal prediction using visual and auditory stimuli. Supplementary Table 1 lists the study coordinates and indicates in which areal pmap of Julich-Brain[128] these coordinates were localised. **E** Functional profiles of PM areas as revealed by the co-localisation of study coordinates visualised in (**A**–**D**) and the respective pmap.

part of Brodmann's area 6[8]. The caudal boundary was given by the already described area 4[114]. The rostral border ventrally to the level of the IFS was characterised by the preCS and the caudal border of area 44 mapped by Amunts et al.[115] and the ifj areas within the junction of the IFS and the preCS[116]. The rostral border dorsally to the level of the IFS was defined by the border of a granular to an (dys-)granular cortex type. The largest portion of this border on the SFG, SFS and MFG has already been identified by the caudal borders of areas 8d1, 8d2, 8v1 and 8v2[111]. The ventral border of the PM was defined by the already mapped areas SMA and pre-SMA by Ruan et al.[117]. The ventral border by areas of the frontal operculum (Op6) mapped by Unger et al.[118].

### Observer-independent border detection
Each ROI was scanned with a resolution of 1.02 μm/pixel using an Axiovision (Zeiss, Germany) connected to a microscope (Axioplan 2 imaging, Zeiss, Germany) and a CCD-Camera (Axiocam MRm, Zeiss, Germany). GLI images were determined by inhouse MatLab scripts for Windows (MatLab R2009a; Mathworks Inc., Natick, MA, USA). The GLI is a measure of the packing density of cell bodies in a field of $16 \times 16$ pixel in the scanned ROI[119], and is correlated to the cell packing density. The localisation of cytoarchitectonic borders between two areas was detected by an observer-independent method using image analysis and multivariate statistics[120]. To characterise laminar packing densities, profiles were extracted from the GLI images using in-house programmed applications of MatLab for Windows (MatLab R2009a; Mathworks Inc., Natick, MA, USA). First, two contour lines were defined. The outer contour was set at the layer I/layer II border and the inner contour at the layer VI/WM border. These two lines were used as start and endpoints of vertically orientated traverses along which GLI

profiles were extracted[121]. Profile shape was quantified by extracting a feature vector with ten elements based on central moments (mean density, mean x, standard deviation, skewness, kurtosis of the profile and the same features of the first derivative). Differences between feature vectors, i.e. in profile shape, were measured using the Mahalanobis Distance. To increase the signal-to-noise ratio, averaged feature vectors of two adjacent blocks of profiles, including 12–24 profiles per block in the GLI images, were calculated. Mahalanobis distance functions were computed by calculating the distances between all pairs of neighbouring cortical blocks using a sliding window approach and plotting the values as a function of the profile position (Fig. 1). Maxima of the Mahalanobis distances revealed the most dissimilar laminar pattern, i.e. a border between areas. The significance ($p < 0.001$) of these maxima was assessed by a Hotelling $T^2$-test and a Bonferroni correction. In summary, significant maxima of the Mahalanobis distance between blocks of density profiles indicated locations of cortical borders.

### Analysis of volumes of areas
Volumes of areas were determined for each hemisphere of the ten brains. The calculation was based on contour lines of the given area drawn on the digitised cell-body stained sections[111]. Furthermore, a shrinkage factor was computed for each brain to correct the areal volumes for volume loss during histological processing. It was determined as a ratio between the fresh volumes and the volumes after fixation[122].

To analyse inter-hemispheric and gender differences of areal volumes, Monte Carlo permutation tests ($p < 0.05$; Bonferroni corrected for multiple comparisons) were carried out using the MatLab Statistics Toolbox (MatLab R2009a; Mathworks Inc., Natick, MA, USA). Thereby, areal volumes were presented as a fraction of whole-brain volume to correct for differences in total brain sizes. Differences were considered to be significant if they were larger than 95% of values under the null-hypothesis ($p < 0.05$, false discovery rate (FDR) corrected for multiple comparisons)[123].

### Cytoarchitectonic areal maps in 3D space
The methodology of reconstructing and calculating areal maps in 3D space has been described in Amunts et al.[111]. The nomenclature of the Julich-Brain atlas has been applied for neighbouring areas and, if not yet mapped for this atlas, Brodmann areas were indicated.

Cytoarchitectonic areas were 3D reconstructed using the three data sets of each of the ten mapped brains: (i) the structural MRI data set of the fixed brain, (ii) the photo data set of blockfaces recorded during sectioning, and (iii) the data set of high-resolution flatbed scans of the cell-body stained sections. The detected cytoarchitectonic borders of the areas were

**Table 3 | 3D correlation of left and right hemispheric areas FEF and PEF of the multimodal map by Glasser et al.[12] and PM areas in ICBM152 space**

| Area in Glasser's map | 3D correlation with PM area (correlation coefficient) |
|---|---|
| FEF L | 6v1 (0.06) |
| FEF R | 6v1 (0.20), 6v2 (0.07) |
| PEF L | 6v2 (0.41), 6v3 (0.24) |
| PEF R | 6v2 (0.44), 6v3 (0.13) |

The correlation of additional areas of the map by Glasser et al. and PM areas are shown in Supplementary Table 2.
*L* left hemisphere, *R* right hemisphere.

**Table 4 | Post mortem brains used for cytoarchitectonic mapping**

| Brain code | Gender | Age | Cutting direction | Mapped areas |
|---|---|---|---|---|
| BC01 | Female | 79 | Coronal | 6d1-3, 6v1-3, 6r1 |
| BC04 | Male | 75 | Coronal | 6d1-3, 6v1-3, 6r1 |
| BC05 | Female | 59 | Coronal | 6d1-3, 6v1-3, 6r1 |
| BC09 | Female | 79 | Coronal | 6d1-3, 6v1-3, 6r1 |
| BC10 | Female | 85 | Coronal | 6d1-3, 6v1-3, 6r1 |
| BC11 | Male | 74 | Coronal | 6d1-3, 6v1-3, 6r1 |
| BC20 | Male | 65 | Coronal | 6d1-3, 6v1-3, 6r1 |
| BC06 | Male | 54 | Coronal | 6d1-3 |
| BC02 | Male | 79 | Coronal | 6d1-3 |
| BC19 | Female | 79 | Sagittal | 6d1-3 |
| BC07 | Male | 37 | Coronal | 6v1-3, 6r1 |
| BC21 | Male | 30 | Coronal | 6v1-3, 6r1 |
| BC08 | Female | 72 | Coronal | 6v1-3, 6r1 |

Each area was mapped in 10 brains (5 male and 5 female brains). The areas were not mapped in the same brains due to artefacts in some regions of interest or for achieving a better cutting angle to the cortical surface by a different cutting direction (e.g. sagittal cut brain used for areas on the dorsal surface).

transferred on scans of the sections using the in-house software *section-tracer*. Both linear and nonlinear transformations were applied to correct for deformations during histological processing. The reconstructed brains and areas were warped to the single-subject brain Colin27 of the MNI reference space and the ICBM152 non-linear asymmetric 2009c[124] by linear and nonlinear elastic transformations. The identified areas were superimposed to generate probability maps[111]. Colour coding in these maps indicates the probability an area occurs in the reference brain. Blue voxels were indicating a high spatial variability, red voxels a low variability, meaning a high probability that these areas can be found in all brains in this voxel. Descriptions such as high and low variability refer to a comparison between the maps of the areas mapped in this study.

The centres of gravity of the areas are based on the pmap of the respective area and are computed using the fslstats program, which is part of the FSL software distribution. fslstats computes the centre of gravity (or centre of mass) based on the voxel intensities in the 3D image, where the intensity of each voxel serves as its weight. It is computed only for voxels that have non-zero probability, i.e., voxels that contribute to the region of interest.

In addition, an MPM of each area was calculated[111,125]. Therefore, the probabilities in each voxel of all areas were compared so that a voxel was assigned to the area which showed the highest probability. Voxels with identical probabilities were assigned to the area with the highest averaged probability of the neighbouring voxels. Border regions where neighbouring areas are not yet mapped were 40% thresholded. In addition, MPMs are computed on the FreeSurfer fsaverage surface[111]. Mapping data were annotated and uploaded to the human brain atlas of the EBRAINS research infrastructure (https://atlases.ebrains.eu) as an open-source tool for the research community.

### Multidimensional scaling and cluster analysis

To visualise cytoarchitectonic similarities and differences of the areas, an MDS analysis was performed. For this purpose, feature vectors were extracted from 45 GLI profiles of three different sections per hemisphere and area, and averaged. The shape, and consequently the descriptive mathematical features, of these profiles characterised and quantified the cytoarchitecture of the underlying brain area and are described by 10 feature vectors. The feature vectors are used to calculate Euclidean distances via MATLAB's pdist function. Subsequently, the mdscale function generates a two-dimensional representation of these distances. The MDS plot serves as a descriptive visualisation of the inter-areal distances. A low Euclidian distance between areas indicates cytoarchitectonic similarities and suggests that these areas belong to a common group. In contrast, a high Euclidean distance specified cytoarchitectonic dissimilarities and suggest different groups. The cluster analysis was also based on the Euclidean distance between the feature vectors of the areas. The unrooted tree was calculated with the program SplitsTree[126] by using the UPGMA (unweighted pair group method with arithmetic mean) as linkage method.

### Comparison of activation coordinates in common stereotaxic space

To visualise activation coordinates reported in functional studies and to compare them with cytoarchitectonic maps in MNI space (ICBM152 non-linear asymmetric 2009c), we used MatLab (MatLab R2009a; Mathworks Inc., Natick, MA, USA), the Statistical Parametric Mapping (SPM12) software package (https://www.fil.ion.ucl.ac.uk/spm/software/spm12/) and an inhouse Matlab script which can be found on GitHub: https://github.com/INM-1-FZJ-Cytoarchitecture/Figure_Template_Brain_with_coords. The coordinates that were originally reported in Talairach space were converted to MNI space (Colin27) using the BioImage Suite Web [33]. Activation coordinates of functional studies plotted in Fig. 6 were selected as follows. The studies and reported coordinates included in Fig. 6A, B were collected from the studies listed and discussed in four review articles by Bedini et al.[7], Vernet et al.[5], Amiez and Petrides[6] and Petit and Pouget[32] on the FEF. The studies used for reaching and grasping in Fig. 6B were taken from an review

article on reaching and grasping actions by Sartin et al.[127]. Chosen were only studies without visual information to exclude FEF activations. Studies providing coordinates for distinct activations in the PM for simple foot/leg and mouth motion, without any language task or similar, were rare in the literature. Therefore, we used those studies that fulfilled this criterion. The studies and coordinates shown in Fig. 6C were chosen because spatial, object-related and temporal prediction was studied in the same set up and the same subjects to reduce the scattering of data based in methodological differences between studies. In addition to the visualisations in Fig. 6, Supplementary Table 1 listed the cytoarchitectonic areas of Julich-Brain (v3.1)[128] whose pmaps contained the corresponding coordinate point. The p-values indicated the probability with which the respective area occurred at this point in the reference brain. This analysis was performed using the assignment function in siibra explorer (https://siibra-explorer.readthedocs.io/en/latest/basics/looking_up_coordinates/). To reduce the complexity, but keep at the same time as much information on interindividual variability as possible, the pmap threshold was set at $p = 0.2$.

### 3D correlation of areas of the multimodal map by Glasser et al. and Julich-Brain areas

To test the spatial similarities of PM areas in the map by Glasser et al.[12] and the PM areas mapped in this study, we calculated the 3D correlation. The volumes of the FEF and PEF areas (https://figshare.com/articles/dataset/HCP-MMP1_0_projected_on_MNI2009a_GM_volumetric_in_NIfTI_format/3501911) were compared to Julich-Brain areas v3.1[128] in ICBM152 2009a space using an in-house script for siibra-python (https://siibra-python.readthedocs.io/en/latest/). The similarity of two areas was quantified by a correlation coefficient. In summary, the correlation coefficient ranged from −1 to +1, whereby +1 expressed a perfect positive correlation-values changed together in exactly the same way throughout the 3D space, 0 meant no correlation—values vary independently, and −1 described a perfect negative correlation—values increased in one dataset, they decrease in the other. In Table 3 and Supplementary Table 2, only correlations of areas with a coefficient of >0.01 were indicated.

### Statistics and reproducibility

Statistical tests were performed on a sample of $n = 10$ brains using MatLab (MatLab R2009a; Mathworks Inc., Natick, MA, USA).

### Reporting summary

Further information on research design is available in the Nature Portfolio Reporting Summary linked to this article.

### Data availability

The probabilistic maps of areas 6d1[129], 6d2[130], 6d3[131], 6v1[132], 6v2[133], 6v3[134] and 6r1[135] are accessible on EBRAINS. The maximum probabilistic map is part of the Julich-Brain Atlas (v.3.1)[128].

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

## Acknowledgements

This work had received funding from the European Union's Horizon 2020 Research and Innovation Programme under Grant Agreement No. 945539 (HBP SGA3), No. 101147319 (EBRAINS 2.0 Project) as well as from the Helmholtz Association's Initiative and Networking Fund through the Helmholtz International BigBrain Analytics and Learning Laboratory (HIBALL) under the Helmholtz International Lab grant agreement InterLabs-0015.

## Author contributions

B.S, J.S. and S.H.R. performed the cytoarchitectonic mapping study. S.H.R. extracted density profiles of areas for MDS and cluster analysis and performed comparisons with functional studies. S.B. performed statistical analyses. H.M. calculated probabilistic maps. S.H.R. prepared the figures and manuscript with assistance of K.A., B.S. and P.P. K.A. and S.C. supervised the research.

## Funding

## Competing interests

The authors declare no competing interests.
