## [Transparent Peer Review file · Communications Biology]

Revised cytoarchitectonic mapping of the human premotor cortex identifies seven areas and refines the localisation of frontal eye fields

Corresponding Author: Dr Sabine Ruland

This manuscript has been previously submitted at another journal. This document only contains information relating to versions considered at Communications Biology.

Version 0:

Reviewer comments:

Reviewer #1

(Remarks to the Author)

Ruland and colleagues use an observer-independent technique to parcellate cytoarchitectonic areas within human premotor cortex, which revealed a novel dorsal-ventral parcellation. Further, the algorithmic approach identified the superior frontal sulcus as a landmark identifying this distinction. The authors also identify initial functional profiles of these cytoarchitectonic areas. I think this a very well-written and comprehensive study. Nevertheless, I do have a major concern below linking the novel cytoarchitectonic parcellation here at the individual level to functional areal parcellations also at the individual level identified previously.

Major Concerns:

1. The authors identify an interesting relationship with this observer-independent approach and cytoarchitectonic boundaries in which the SFS identifies a landmark. Looking at their data, I believe there is an additional landmark. Specifically, Mackey and colleagues (2017, eLife) identified that the intersection between the posterior portion of the SFS and the superior portion of the precentral sulcus identified an eccentricity cluster and two visual field maps (their Figure 4). This seems to be the same anatomical landmark in the present data delineating the boundary between areas 6d1 and 6d2. And further, the areas (SPCS1 and SPCS2) identified by Mackey and colleagues also have a dorsal-ventral arrangement which could further relate to a boundary between 6d1 and 6v1 identified by the present authors. Finally, based on the location of the area IPCS relative to the posterior extent of the IFS by Mackey and colleagues, it looks like it could be what the present authors identify as 6v3. It would be helpful for the authors to use the sulcal patterning to map these areas between studies as the correspondence seems very strong. The data are freely available from the Mackey et al. study. The correspondence between studies seems quite remarkable relative to sulcal patterning at the level of individual hemispheres - which is a more of a "like vs. like" comparison compared to the functional profiles identified in the current version which is at the meta-analytic level in stereotaxic coordinates.

Reviewer #2

(Remarks to the Author)

The study "Decoding the human premotor cortex: organisational principles, 3D maps and functional relationships" by Ruland et al. investigates the microstructural organization of the human premotor cortex (PM) using an observer-independent cytoarchitectonic mapping approach in ten post-mortem brains. Seven distinct PM areas (6d1-6d3, 6v1-6v3, 6r1) are identified and grouped into dorsal and ventral premotor subdivisions. The authors provide probabilistic maps in 3D space and demonstrate structure-function relationships by comparing these maps with functional imaging data. Notably, the study clarifies the localization of the frontal and inferior frontal eye fields within the PM, rather than the prefrontal cortex. The new maps are made publicly available, providing a crucial resource for future neuroimaging research. This study provides a significant advancement in understanding the cytoarchitectonic organization of the premotor cortex and its relationship with functional activations. This manuscript should be published in Communications Biology; however, some areas require

clarification, restructuring, and further discussion before acceptance.

1. Overall, the introduction is comprehensive but could be streamlined in some areas. For example, the discussion of historical PM maps at the beginning could be condensed to focus on the most relevant studies. Summarizing these previous maps in a table would help clarify and also provide a good historical resource.
2. The methods section is quite thorough but could benefit from clearer subheadings and a more structured breakdown of each methodological step, particularly in the "Cytoarchitectonic Mapping" subsection.
3. The study is based on ten post-mortem brains, which is a reasonable sample for histological studies, but there is no discussion on potential limitations due to interindividual variability. How this sample size impacts the generalizability of the findings should be discussed.
4. The clustering of PM areas is convincing, but a clearer rationale for using this technique is needed. Additionally, could alternative clustering methods be used to further validate these findings?
5. The study references various sulcal landmarks (e.g., precentral sulcus, superior frontal sulcus), but the reliability of these as anatomical borders is not sufficiently validated. Given the notable interindividual variability in prefrontal sulci, as well as their anatomical, cognitive, and functional significance (e.g., Amiez & Petrides, 2018 *Brain Structure & Function*; Miller et al., 2021 *Journal of Neuroscience*; Willbrand et al., 2023 *Journal of Neuroscience*) the authors should provide a more detailed analysis of interindividual variability in sulcal morphology and its impact on defining cytoarchitectonic borders. Further, do features of these sulci (eg, depth) impact the cytoarchitectonic borders?
6. The maps of all postmortem brains should be included at least in the supplements.
7. Aligning the PM borders to other template spaces (eg, fsaverage) could be useful for researchers using other templates and to further assess the consistency of the borders.

Reviewer #3

(Remarks to the Author)

The paper of Ruland and colleagues makes a highly relevant and much-needed contribution to the ongoing debate on the subfield organization of the lateral premotor cortex in the human brain, including the frontal eye field. Through an observer-independent analysis of the cytoarchitectonic properties of the premotor cortex, the authors identify seven distinct areas, three of which cluster as 'dorsal' and four as 'ventral' lateral premotor fields. Notably, the superior frontal sulcus emerges as a key macroanatomical landmark distinguishing these two subdivisions. In addition, the authors propose functional characteristics for these premotor subfields by integrating findings from functional MRI studies on motor and cognitive functions in the lateral premotor region. Finally, based on functional considerations, they suggest that two of the ventral areas correspond to the frontal eye field and the inferior frontal eye field.

This study represents a carefully conducted and highly valuable contribution to the field. We welcome it as a new reference for lateral premotor parcellation based on microstructural features. The authors' rigorous methodology and clear anatomical delineation provide a strong foundation for future research on the functional organization of the premotor cortex and a better understanding of its structural and functional heterogeneity.

While the paper is already of high quality, I have a few suggestions that may further enhance its clarity, precision, and broader impact. I encourage the authors to consider the following points to strengthen their manuscript.

1. While the cytoarchitectonic profiling of PM areas was conducted in an observer-independent manner, this does not appear to be the case for the description of topography and its relationship to macroanatomical landmarks—if I have understood correctly. If this aspect was indeed not observer-independent, it would be helpful for the authors to explain why this approach was taken, whether alternative methods are available, and if this represents a potential limitation. At the very least, making this point explicit would enhance clarity.
2. The authors use MDS to reveal clusters, but it remains unclear whether this is based on an actual cluster analysis within the MDS space or if it is intended as a more descriptive approach. Clarifying this distinction would greatly enhance the reader's understanding of the methodological framework. Additionally, more details on how the MDS analysis was conducted would be beneficial. In particular, the distinction between 'high' and 'low' Euclidean distances between areas appears to serve as a somewhat soft marker rather than a clearly defined criterion. It would be helpful if the authors could elaborate on how these thresholds were determined and whether they represent an objective measure within the clustering process.
3. Figure 2 part C: Could the authors clarify whether the brains used for subfield identification in this study are the same as those used to generate the probabilistic maps of the lateral premotor cortex? The methods seem to imply this, yet the apparent size difference in the figure raises some questions. If the same brains were used, what might explain the discrepancy, particularly the fact that the 10 brains in this study did not reach the anterior boundary of the probabilistic field size? Providing a brief explanation would help clarify this aspect.
4. In Figure 2B, the cytoarchitectonic probability maps of areas 6d3 and 6v2 are projected onto the surface of the stereotaxic MNI Colin27 template. However, for a more comprehensive overview, it would be valuable to see the probability maps for all seven areas identified in this study. Including these maps would provide a clearer representation of the full parcellation and enhance the completeness of the figure.

5. Could the authors clarify how the centers of gravity for the cytoarchitectonic probabilistic maps were calculated? It is possible that I may have overlooked this information in the manuscript, but if it is indeed provided, I would still recommend including a brief mention in the legend of Table 1 for clarity and ease of reference.
6. Interestingly, both dorsal and ventral premotor areas appear to show a comparable similarity to opercular areas, as illustrated in Figure 3. How can this be explained? Conversely, if proximity in the MDS space is a critical feature for classifying some areas as part of the ventral premotor cortex and others as part of the dorsal premotor cortex, does it not pose a problem that dorsal premotor regions seem just as close to the opercular areas as the ventral premotor regions? Given that Sanides' gradations (1962) suggest that ventral premotor areas should be more similar to opercular regions than dorsal premotor areas, this pattern appears somewhat unexpected. At least from the figure, this seems to be the case, though I may be overlooking some details. Some further clarification on this point would be helpful.
7. The authors refer to 30 studies reporting activity coordinates for the FEF. Could the authors clarify the selection criteria for these studies? Were they included based on functional relevance, i.e., only studies that explicitly tested eye movements, or was the selection primarily based on the use of the label 'FEF' in those studies, assuming it to be the most appropriate designation given prior research? In this context, I also noticed that while task and experimental condition descriptions are provided for Figures 5C and 5D, they are missing for Figures 5A (FEF) and 5B (iFEF) in Supplementary Table 1. To ensure completeness and clarity, it would be important to include these descriptions as well.
8. The authors mention other approaches to the parcellation of the lateral premotor cortex, particularly those by Glasser et al. (2016) and Fan et al. (2016). However, their comparison is limited to only the PEF and FEF regions from Glasser's study, while the remaining regions identified by Glasser et al. (6d, 6a, 55b, 6v, and 6r), and the parcellation of Fan et al. (6dl, 6cdl, 6vl, and 6cvl) are not addressed. This selective comparison may result in an incomplete evaluation of the existing parcellation frameworks. To provide a more comprehensive analysis, I recommend extending the comparison to include all regions of the premotor cortex (or BA6) reported by both Glasser et al. and Fan et al. This would offer a more balanced assessment of these other parcellation approaches and help clarify the relationship between the studies. Additionally, the authors may consider commenting on the discrepancies between Glasser et al.'s PEF and FEF coordinates and their own findings. These differences are interesting and could be discussed a bit more detailed to better understand how the findings align or diverge. Finally, while the authors mention that the differences in parcellation could be attributed to the different analysis pipelines used by Glasser and Fan, I believe it would be valuable to explore both parcellations in a more detailed manner. As I said, including all seven areas from Glasser and four areas from Fan would strengthen the comparison and provide a clearer context for understanding the observed differences.
9. I would appreciate a more detailed description of the specific region analyzed in the brains and the criteria used for defining it. Currently, the information provided (l. 494-495) states that "Regions of interest (ROIs) were identified in images of histological sections which encompassed the PM and adjoining areas. The caudal boundary was given by the already described area 4 [35]," but this only clarifies the caudal boundary. It would be helpful to also specify the rostral, ventral, and dorsal boundaries. Additionally, it is unclear whether "PM" refers specifically to BA 6. Clarifying these aspects would improve transparency and reproducibility.
10. Premotor subfields have been a focus of research for several decades, and it would be valuable to connect the present findings in humans, at least to some extent, with this body of work. The authors could consider discussing how the seven subfields correspond to the macaque's rostroventral (F5), caudoventral (F4), rostradorsal (F7), and caudodorsal (F2) fields as described by Geyer et al. (2000). Furthermore, the authors may reflect on the additional subdivisions of F5 (and possibly other areas) that have emerged in more recent macaque research.
11. As a personal sidenote, twenty years ago in my habilitation, I summarized the discussion on the ventral/dorsal boundary of the human lateral premotor cortex as follows: "Homologies between human and monkey PMv and PMd are especially difficult to determine. This is partly due to the fact that in the monkey, cytoarchitectonic and microstimulation studies have provided conflicting evidence on the ventral-dorsal subdivision of the premotor cortex. The PMd-PMv boundary has been attributed either to the spur of the arcuate sulcus of the macaque (Rizzolatti et al., 1998, 2002) or to the inferior arcuate dimple in the owl monkey, possibly corresponding to the inferior precentral dimple of the macaque (Preuss et al., 1996). [...] According to Preuss, the human homologue of caudal PMd occupies the precentral gyrus (corresponding to area 6α), whereas that of rostral PMd and area 8b occupies the dorsal frontal cortex rostral to the precentral sulcus (corresponding to area 6β). Since both monkey PMv and human BA 44 are dysgranular (Bucy, 1944), and both monkey PMv as well as human BA 44 and/or ventral BA 6 represent upper limb and orofacial movements, Preuss proposes the human homologue of PMv to correspond to area 44 and the ventral part of area 6. In contrast, for Rizzolatti and co-workers (Rizzolatti et al., 1998, 2002) one important consideration is that during ontogenesis, human superior and inferior precentral sulcus develop from two separate primordia as vertical branches of the superior and inferior frontal sulcus (Turner, 1948). In view of functional differences between ventral and dorsal premotor cortex, it would be plausible to suggest this dual origin to be reflected in parallel functional differences. A crucial assumption here is that the functional areas delimited by the most ancient sulci maintain their basic location in phylogeny. This would imply that the human homologue of the superior arcuate sulcus is the superior precentral sulcus plus superior frontal sulcus. Then, dorsal area 6α and 6β would correspond to F2 and F7, respectively (cf. Zilles et al., 1995). The human homologue of the inferior arcuate sulcus would be the ascending branch of inferior precentral sulcus plus inferior frontal sulcus. Finally, the descending branch of inferior precentral sulcus in humans would be equivalent to the inferior precentral dimple in the monkey, and hence, human ventral area 6α and area 44 would correspond to F4 and F5 respectively. Rizzolatti and co-workers therewith propose that human PMd is located superiorly and PMv inferiorly to about z=51 of Talairach space." So, in the meantime, many more studies have been conducted on this topic, and while not all details may still be fully endorsed, it is nevertheless remarkable that the authors'

study places the boundary between the ventral and dorsal lateral premotor cortex at the level of the extension of the SFS (superior frontal sulcus), in accordance with this argumentation. This is a very nice convergence!

Minor points:

Figure 1, part A, right hand side: "76" seems to be cut off

Typos: sometimes "figure X" instead of „Figure X“

l. 57: „Fan et al. [13]“ should read „Fan et al. [71]“ I think

Sex of donors of the brains was not mentioned

It would be helpful to explain the abbreviations used, such as 'd' for dorsal, 'l' for lateral, 'c' for caudal, 'r' for rostral etc. (e.g., in 6dl, 6cdl, 6r). Providing this explanation once at the beginning would be sufficient.

l. 214-215 "The ventral border was identified dorsally to the LF". Please add the ventrally bordering of area Op6 (frontal operculum), which is important to avoid a direct border with the lateral fissure. This is a particularly relevant clarification for those who don't know that in the human precentral gyrus, the most ventral portion is not corresponding to BA 6 but to the frontal operculum.

Figure Legend 2: „lateral fissure preCS“ should read „lateral fissure; preCS“

l. 319-320: Regarding this point, it may be worth reconsidering the label 'attention' for the serial prediction tasks reported by Schubotz. While attention can also elicit activity in premotor areas, the premotor activations in these studies were found specifically when contrasting predictive tasks on stimulus sequences with equally attentionally demanding but non-predictive tasks (e.g., serial match-to-sample). Since attention and working memory were controlled for, they are unlikely to fully explain the observed effects. Instead, predictive task instructions on stimulus sequences or dynamic stimuli appear to be the primary drivers of premotor activity, as demonstrated across a series of studies. The authors may find a useful summary in Schubotz, 2007, Trends in Cognitive Sciences.

Regarding the discussion of the FEF, authors may take into account the work of Tehovnik et al., 2000 (Tehovnik, E., Sommer, M., Chou, I., Slocum, W. & Schiller, P. (2000). Eye fields in the frontal lobes of primates. Brain Res Brain Res Rev, 32(2-3), 413–48) and Preuss et al 1996 (Preuss, T., Stepniewska, I. & Kaas, J. (1996). Movement representation in the dorsal and ventral premotor areas of owl monkeys: a microstimulation study. J Comp Neurol, 371(4), 649–76).

l. 401-402 state: "The interindividual variability in sulcal pattern, which is particularly high for the region of the junction of the preCS and the SFS." Could you clarify what this variability is being compared to? In other words, what criterion is used to classify an anatomical landmark as exhibiting "low" versus "high" variability? Providing this context would help in better understanding the statement.

l. 435-437: „These results align with monkey studies, which show overlapping representations of the hindlimb and arm in the PMd, and of the forelimb and face representation within the PMv [92]. Human functional studies have also provided evidence of a similar organisation within the human PM (e.g. [2, 93-95]).“ Authors may also consider (and possibly cite) Sakreida, K., Schubotz, R.I., Wolfensteller, U., von Cramon, D.Y. (2005). Motion class dependency in observers' motor areas revealed by functional MRI. The Journal of Neuroscience, 25(6), 1335-42.

Version 1:

Reviewer comments:

Reviewer #1

(Remarks to the Author)

The authors have addressed each of my concerns and I have no further comments. It is a suiting contribution to Communications Biology.

Reviewer #2

(Remarks to the Author)

I believe that the authors have adequately addressed all of my comments and believe the paper is ready to be published.

Reviewer #3

(Remarks to the Author)

I appreciate the authors' thorough responses to my comments. The paper stands out as a strong, engaging, and methodologically ambitious contribution. Congratulations!

Response to referees

We, the authors, sincerely thank the reviewers for taking the time to carefully evaluate our manuscript and for providing valuable feedback and constructive comments that helped us improve the clarity, accuracy, and overall quality of the work.

Below, we provide a detailed point-by-point response to each comment. For each point, we indicate how and where the suggested changes have been addressed in the revised manuscript (*responses and manuscript changes are indicated in italics*).

Reviewer #1 (Remarks to the Author):

Ruland and colleagues use an observer-independent technique to parcellate cytoarchitectonic areas within human premotor cortex, which revealed a novel dorsal-ventral parcellation. Further, the algorithmic approach identified the superior frontal sulcus as a landmark identifying this distinction. The authors also identify initial functional profiles of these cytoarchitectonic areas. I think this a very well-written and comprehensive study. Nevertheless, I do have a major concern below linking the novel cytoarchitectonic parcellation here at the individual level to functional areal parcellations also at the individual level identified previously.

Major Concerns:

1. The authors identify an interesting relationship with this observer-independent approach and cytoarchitectonic boundaries in which the SFS identifies a landmark. Looking at their data, I believe there is an additional landmark. Specifically, Mackey and colleagues (2017, eLife) identified that the intersection between the posterior portion of the SFS and the superior portion of the precentral sulcus identified an eccentricity cluster and two visual field maps (their Figure 4). This seems to be the same anatomical landmark in the present data delineating the boundary between areas 6d1 and 6d2. And further, the areas (SPCS1 and SPCS2) identified by Mackey and colleagues also have a dorsal-ventral arrangement which could further relate to a boundary between 6d1 and 6v1 identified by the present authors. Finally, based on the location of the area IPCS relative to the posterior extent of the IFS by Mackey and colleagues, it looks like it could be what the present authors identify as 6v3. It would be helpful for the authors to use the sulcal patterning to map these areas between studies as the correspondence seems very strong. The data are freely available from the Mackey et al. study. The correspondence between studies seems quite remarkable relative to sulcal patterning at the level of individual hemispheres - which is a more of a "like vs. like" comparison compared to the functional profiles identified in the current version which is at the meta-analytic level in stereotaxic coordinates.

Thank you very much for this important comment. The study by Mackey et al. supports our finding that the SFS acts as a reliable anatomical landmark for functional subdivisions within the PM. In their work, the SFS separates two visual field maps, sPCS1 and sPCS2.

Importantly, Mackey et al. report that within sPCS, the foveal representation is located in the fundus of the preCS at its intersection with the SFS. A comparison of the individual maps and sulcal patterns shown in their Figure 4 and Supplementary Figure 1 with our mappings (Figure 2) suggests that this point corresponds spatially to the intersection of areas 6d1,

6d2/6d3, and 6v1 within the SFS in our data. Notably, sPCS2 may align with 6v1 in our study (which shows a correlation with the FEF area of Glasser et al. 2016), while sPCS1 may correspond to 6d1 and 6d3, and possibly parts of 6d2.

Regarding the ventrally located iPCS, the foveal representation was found at the intersection of the IFS and the preCS. This anatomical location matches the location of the inferior frontal junction (ifj) areas (ifj1, ifj2) mapped in Ruland et al. 2022 in Cortex. However, comparing the location of iPCS with our map is more challenging due to the reported variability across subjects in that region ("the topography in the iPCS region was not sufficiently regular across subjects"). According to Figure 8 and Supplementary Figure 2 in Mackey et al., iPCS may encompass areas ifj1, ifj2 and 6v3, as well as ventral parts of 6v2 and 6r1.

Overall, the description of the two clusters - sPCS and iPCS - suggests a strong anatomical and functional relationship between sPCS and iPCS and the two eye fields – FEF and iFEF – as identified in functional studies.

We have now included the study by Mackey et al. in the discussion. See lines 642-644 in the manuscript.

Reviewer #2 (Remarks to the Author):

The study "Decoding the human premotor cortex: organisational principles, 3D maps and functional relationships" by Ruland et al. investigates the microstructural organization of the human premotor cortex (PM) using an observer-independent cytoarchitectonic mapping approach in ten post-mortem brains. Seven distinct PM areas (6d1-6d3, 6v1-6v3, 6r1) are identified and grouped into dorsal and ventral premotor subdivisions. The authors provide probabilistic maps in 3D space and demonstrate structure-function relationships by comparing these maps with functional imaging data. Notably, the study clarifies the localization of the frontal and inferior frontal eye fields within the PM, rather than the prefrontal cortex. The new maps are made publicly available, providing a crucial resource for future neuroimaging research. This study provides a significant advancement in understanding the cytoarchitectonic organization of the premotor cortex and its relationship with functional activations. This manuscript should be published in *Communications Biology*; however, some areas require clarification, restructuring, and further discussion before acceptance.

1. Overall, the introduction is comprehensive but could be streamlined in some areas. For example, the discussion of historical PM maps at the beginning could be condensed to focus on the most relevant studies. Summarizing these previous maps in a table would help clarify and also provide a good historical resource.

We included a table (Tab. 1) in the introduction summarising the different parcellation schemes. See lines 66-74 in the manuscript.

2. The methods section is quite thorough but could benefit from clearer subheadings and a more structured breakdown of each methodological step, particularly in the "Cytoarchitectonic Mapping" subsection.

We introduced additional subheadings in the methods part. See lines 712 (Histological processing), 734 (Identification of Region of Interest (ROI)), and 747 (Observer-independent border detection) in the manuscript.

3. The study is based on ten post-mortem brains, which is a reasonable sample for histological studies, but there is no discussion on potential limitations due to interindividual variability. How this sample size impacts the generalizability of the findings should be discussed.

The sample size of post mortem studies is typically smaller than in in vivo neuroimaging studies due to the time- and labour-intensive tissue preparation, staining and analysis. This limitation affects the statistical power of post mortem studies, which tends to be lower than those in MR studies (Meurs, 2016, Forensic Sci Med Pathol), particularly when analysing subgroup differences such as hemispheric asymmetries or gender effects - where group size may drop to as few as five brains. Consequently, only large differences reach statistical significance. Despite this limitation, post mortem microstructural studies offer distinct advantages, particularly in terms of spatial resolution and the precision of defining microstructural borders between cortical areas. This enables a highly detailed analysis of relationship between microstructural borders and individual sulcal patterns, contributing to our understanding of intersubject variability. The resulting probabilistic maps capture and quantify this variability. Based on our experience, these maps remain robust—even when individual post mortem brains are replaced—indicating that the spatial distribution of cortical areas remains stable across subjects. The maps serve as a valuable resource for estimating the probability that a given area is located at a specific position in the brain. Importantly, the maps are integrated in the EBRAINS Infrastructure, providing a framework for the integration of additional datasets and cohorts.

We included a paragraph on this aspect in the discussion, see lines 559-567.

4. The clustering of PM areas is convincing, but a clearer rationale for using this technique is needed. Additionally, could alternative clustering methods be used to further validate these findings?

The MDS analysis was conducted using a Euclidean distance matrix, calculates from the aggregated predictor variables, which were measured across three histological sections per area and hemisphere. These predictor variables (feature vector described in methods) were used to compute Euclidean distances via MATLAB's pdist function. Subsequently, the mdscale function generates a two-dimensional representation of these distances. Importantly, no clustering algorithm is applied within the MDS space; the resulting plot is purely descriptive and intended to visualise the relative distances and complex relationships among areas. The terms "high" and "low" Euclidean distances in our description are used comparatively and do not imply fixed thresholds or discrete clusters within the MDS procedure. This clarification has been added to the Methods section, see lines 841-844. In addition, we have now included a hierarchical cluster analysis, displayed as an unrooted tree (Figure 4B, lines 366-370). The tree was calculated with the program SplitsTree by using the UPGMA (unweighted pair group method with arithmetic mean) as linkage method. This addition is now described in the Methods section (lines 839-853) and the Results (lines 356-369).

5. The study references various sulcal landmarks (e.g., precentral sulcus, superior frontal sulcus), but the reliability of these as anatomical borders is not sufficiently validated. Given the notable interindividual variability in prefrontal sulci, as well as their anatomical, cognitive, and functional significance (e.g., Amiez & Petrides, 2018 *Brain Structure & Function*; Miller et al., 2021 *Journal of Neuroscience*; Willbrand et al., 2023 *Journal of Neuroscience*) the authors should provide a more detailed analysis of interindividual variability in sulcal morphology and its impact on defining cytoarchitectonic borders. Further, do features of these sulci (eg, depth) impact the cytoarchitectonic borders?

Detailed studies on the sulcal variability and probability maps of the SFS, the preCS and IFS have been published previously by Drudik and Petrides, 2024, Human Brain Mapping, Germann et al., 2005, J Comp Neurol. and Nolan et al., 2024, Human Brain Mapping based on high-sample-size MRI datasets. The first two studies are referenced in the present work. In our manuscript, we expand on sulcal-area relationship in greater detail in the Results section "Topography and relationship to macroanatomical landmarks" (starting at line 181). To further illustrate the localisation of areas in relation to sulcal landmarks, we have added individual surface maps of 3 additional brains in Figure 2 as well as coronal/sagittal sections from the remaining brains in Supplementary Figure 2, highlighting areas and borders buried with the sulci.

6. The maps of all postmortem brains should be included at least in the supplements.

We have included now surface maps of 3 additional individual brains in Figure 2. In addition, we provided now coronal and sagittal sections of the remaining brains in Supplementary Figure 2 to illustrate the localisation of areas and their borders buried within sulci as well as their individual variability.

7. Aligning the PM borders to other template spaces (eg, fsaverage) could be useful for researchers using other templates and to further assess the consistency of the borders.

The maps are available in Colin27, MNI152 and fsaverage on EBRAINS (Julich-Brain Atlas v3.1, DOI: 10.25493/KNSN-XB4). This availability is now also emphasized in the legend of Figure 3.

Reviewer #3 (Remarks to the Author):

The paper of Ruland and colleagues makes a highly relevant and much-needed contribution to the ongoing debate on the subfield organization of the lateral premotor cortex in the human brain, including the frontal eye field. Through an observer-independent analysis of the cytoarchitectonic properties of the premotor cortex, the authors identify seven distinct areas, three of which cluster as 'dorsal' and four as 'ventral' lateral premotor fields. Notably, the superior frontal sulcus emerges as a key macroanatomical landmark distinguishing these two subdivisions. In addition, the authors propose functional characteristics for these premotor subfields by integrating findings from functional MRI studies on motor and cognitive functions in the lateral premotor region. Finally, based on functional

considerations, they suggest that two of the ventral areas correspond to the frontal eye field and the inferior frontal eye field.

This study represents a carefully conducted and highly valuable contribution to the field. We welcome it as a new reference for lateral premotor parcellation based on microstructural features. The authors' rigorous methodology and clear anatomical delineation provide a strong foundation for future research on the functional organization of the premotor cortex and a better understanding of its structural and functional heterogeneity.

Thank you very much for this positive feedback!

While the paper is already of high quality, I have a few suggestions that may further enhance its clarity, precision, and broader impact. I encourage the authors to consider the following points to strengthen their manuscript.

1. While the cytoarchitectonic profiling of PM areas was conducted in an observer-independent manner, this does not appear to be the case for the description of topography and its relationship to macroanatomical landmarks—if I have understood correctly. If this aspect was indeed not observer-independent, it would be helpful for the authors to explain why this approach was taken, whether alternative methods are available, and if this represents a potential limitation. At the very least, making this point explicit would enhance clarity

The macroanatomical landmarks (sulci, gyri) of the individual post mortem brains were identified according to Ono et al. 1990, a widely used reference for sulcal identification (see e.g. Juch et al. 2005 NeuroImage). The relationship between sulci and specific areas – such as the CS and area 4, or the IFG, preCS, and area 4 - have been previously described and mapped in these brains in earlier studies. Furthermore, Wang et al. 2024 (in Cerebral Cortex) recently published an automatic identification of cortical folding landmarks in the some of the used brains using BrainVISA, further supporting the anatomical consistency and reproducibility of sulcal identification in this dataset.

2. The authors use MDS to reveal clusters, but it remains unclear whether this is based on an actual cluster analysis within the MDS space or if it is intended as a more descriptive approach. Clarifying this distinction would greatly enhance the reader's understanding of the methodological framework. Additionally, more details on how the MDS analysis was conducted would be beneficial. In particular, the distinction between 'high' and 'low' Euclidean distances between areas appears to serve as a somewhat soft marker rather than a clearly defined criterion. It would be helpful if the authors could elaborate on how these thresholds were determined and whether they represent an objective measure within the clustering process.

The MDS analysis was conducted using a Euclidean distance matrix, calculates from the aggregated predictor variables, which were measured across three histological sections per area and hemisphere. These predictor variables (feature vector described in methods) were used to compute Euclidean distances via MATLAB's pdist function. Subsequently, the mdscale function generates a two-dimensional representation of these distances. Importantly, no clustering algorithm is applied within the MDS space; the resulting plot is

purely descriptive and intended to visualise the relative distances and complex relationships among areas. The terms “high” and “low” Euclidean distances in our description are used comparatively and do not imply fixed thresholds or discrete clusters within the MDS procedure. This clarification has been added to the Methods section, see lines 841-844. In addition, we have now included a hierarchical cluster analysis, displayed as an unrooted tree (Figure 4B, lines 366-370). The tree was calculated with the program SplitsTree by using the UPGMA (unweighted pair group method with arithmetic mean) as linkage method. This addition is now described in the Methods section (lines 839-853) and the Results (lines 356-369).

3. Figure 2 part C: Could the authors clarify whether the brains used for subfield identification in this study are the same as those used to generate the probabilistic maps of the lateral premotor cortex? The methods seem to imply this, yet the apparent size difference in the figure raises some questions. If the same brains were used, what might explain the discrepancy, particularly the fact that the 10 brains in this study did not reach the anterior boundary of the probabilistic field size? Providing a brief explanation would help clarify this aspect.

Figure 2 now shows the mapped areas as surface representations within 6 out of the 10 individual brains. In addition, Supplementary Figure 2 includes the remaining mapped brains, providing full coverage of the dataset. The brains of these two figures form the basis for the calculation of the pmaps and MPM. An overview of the brain sample and mapped areas in each is now given in Table 4 (line 726).

During revision - and thanks to your helpful observation - we identified an issue in the MPM of areas 6d2 and 6d3, particularly in the right hemisphere, where the rostral extent was indeed too pronounced. This has been corrected, and the updated version is now reflected in Figure 3B.

4. In Figure 2B, the cytoarchitectonic probability maps of areas 6d3 and 6v2 are projected onto the surface of the stereotaxic MNI Colin27 template. However, for a more comprehensive overview, it would be valuable to see the probability maps for all seven areas identified in this study. Including these maps would provide a clearer representation of the full parcellation and enhance the completeness of the figure.

We have now included the pmaps of all areas in Figure 3 to provide a comprehensive overview.

5. Could the authors clarify how the centers of gravity for the cytoarchitectonic probabilistic maps were calculated? It is possible that I may have overlooked this information in the manuscript, but if it is indeed provided, I would still recommend including a brief mention in the legend of Table 1 for clarity and ease of reference.

*The centres of gravity of each area were derived from their respective probabilistic maps and calculated using the *fsLstats* tool from the FSL software suite. *fsLstats* computes the centre of gravity (or centre of mass) based on the voxel intensities in the 3D image, with each voxel's intensity serving as its weight. The calculation includes only those voxels with non-zero probability values—i.e., those contributing to the region of interest. This description has also been added to the Methods section (lines 825-829).*

6. Interestingly, both dorsal and ventral premotor areas appear to show a comparable similarity to opercular areas, as illustrated in Figure 3. How can this be explained? Conversely, if proximity in the MDS space is a critical feature for classifying some areas as part of the ventral premotor cortex and others as part of the dorsal premotor cortex, does it not pose a problem that dorsal premotor regions seem just as close to the opercular areas as the ventral premotor regions? Given that Sanides' gradations (1962) suggest that ventral premotor areas should be more similar to opercular regions than dorsal premotor areas, this pattern appears somewhat unexpected. At least from the figure, this seems to be the case, though I may be overlooking some details. Some further clarification on this point would be helpful.

We agree that this line of reasoning is generally sound. Sanides' gradations were based on myelo- and cytoarchitectonic observations, with the cytoarchitectonic gradations primarily reflecting the number and degree of development of cortical layers—what he referred to as “zunehmende und abnehmende Schichtenbetonung” (increasing and decreasing laminar emphasis). However, Sanides did not quantify the full extent of cytoarchitectonic differences. Such quantification has only become feasible in recent years with the development of advanced histological and computational techniques. The quantitative method used in the present study captures a broader and more detailed spectrum of microstructural information. It is based on feature vectors that describe the entire cortical architecture using parameters derived from the shape of the GLI (gray level index) profile curves across cortical layers. Specifically, 10 numerical features characterize the cytoarchitectonic profile of each area, providing a standardized but inevitably reduced representation of the underlying complexity.

Multidimensional scaling (MDS), as applied here, is a statistical technique that allows the visualization of relationships (in terms of similarity or dissimilarity) among data points in a lower-dimensional space. It is well-suited for capturing and visualizing structural patterns in high-dimensional data. In our case, it helps illustrate how areas group together based on their cytoarchitectonic similarity. In previous studies, MDS and cluster analysis have successfully revealed clusters of cortical areas with similar microstructural features. This is also evident in the present dataset: for example, areas of the premotor cortex (PM), dorsolateral prefrontal cortex (DLPFC), and frontal operculum each form distinct and interpretable clusters. However, it is important to acknowledge the limitations of MDS visualization. While the technique aims to preserve the distances between data points as faithfully as possible, the 2D representation inevitably introduces distortions—especially as the number of areas increases. With 18 areas included in the present analysis, not all subtle relationships and gradients can be fully visualized or interpreted in two dimensions.

*Regarding cytoarchitectonic granularity, our results reveal a gradual transition between agranular and (dys)granular cortex within the mapped regions of the PM and adjacent frontal operculum. For example, area 6v3 lacks a layer IV and is therefore agranular, while area 6r1 exhibits a thin, discontinuous layer IV, making it more similar to area 44 and the neighboring opercular regions. The opercular areas themselves have been characterized as dysgranular or granular, as described in Unger et al. (2023, *Frontiers in Human Neuroscience*).*

7. The authors refer to 30 studies reporting activity coordinates for the FEF. Could the authors clarify the selection criteria for these studies? Were they included based on functional relevance, i.e., only studies that explicitly tested eye movements, or was the selection primarily based on the use of the label 'FEF' in those studies, assuming it to be the most appropriate designation given prior research? In this context, I also noticed that while task and experimental condition descriptions are provided for Figures 5C and 5D, they are missing for Figures 5A (FEF) and 5B (iFEF) in Supplementary Table 1. To ensure completeness and clarity, it would be important to include these descriptions as well.

The studies and reported coordinates shown in Figure 6A (previously Figure 5A) were drawn from those reviewed and discussed in four key review articles on the frontal eye fields (FEF): Bedini et al. (2021), Vernet et al. (2014), Amiez and Petrides (2009), and Petit and Pouget (2019). These references are cited in the Methods section (lines 862-878). To enhance clarity and transparency, we have now included the experimental conditions associated with the studies represented in Figures 6A and 6B in Supplementary Table 1.

8. The authors mention other approaches to the parcellation of the lateral premotor cortex, particularly those by Glasser et al. (2016) and Fan et al. (2016). However, their comparison is limited to only the PEF and FEF regions from Glasser's study, while the remaining regions identified by Glasser et al. (6d, 6a, 55b, 6v, and 6r), and the parcellation of Fan et al. (6dl, 6cdl, 6vl, and 6cvl) are not addressed. This selective comparison may result in an incomplete evaluation of the existing parcellation frameworks. To provide a more comprehensive analysis, I recommend extending the comparison to include all regions of the premotor cortex (or BA6) reported by both Glasser et al. and Fan et al. This would offer a more balanced assessment of these other parcellation approaches and help clarify the relationship between the studies. Additionally, the authors may consider commenting on the discrepancies between Glasser et al.'s PEF and FEF coordinates and their own findings. These differences are interesting and could be discussed a bit more detailed to better understand how the findings align or diverge. Finally, while the authors mention that the differences in parcellation could be attributed to the different analysis pipelines used by Glasser and Fan, I believe it would be valuable to explore both parcellations in a more detailed manner. As I said, including all seven areas from Glasser and four areas from Fan would strengthen the comparison and provide a clearer context for understanding the observed differences.

Thank you for commenting on this important aspect. We have now included a comparison with the multimodal cortical parcellation by Glasser et al. (2016) and Fan et al. in the Discussion section (lines 519-546). To further support this comparison, we added Supplementary Table 2, which shows the correlation between our mapped PM areas and the areas 6ma, 6mp, 6a, 6d, 55b, 6v, and 6r in the left and right hemispheres as defined in the multimodal map by Glasser et al.

9. I would appreciate a more detailed description of the specific region analyzed in the brains and the criteria used for defining it. Currently, the information provided (l. 494-495) states that "Regions of interest (ROIs) were identified in images of histological sections which encompassed the PM and adjoining areas. The caudal boundary was given by the already described area 4 [35]," but this only clarifies the caudal boundary. It would be helpful to also specify the rostral, ventral, and dorsal boundaries. Additionally, it is unclear

whether "PM" refers specifically to BA 6. Clarifying these aspects would improve transparency and reproducibility.

We added additional information in the Methods section, Identification of Region of interest (lines 737-749).

10. Premotor subfields have been a focus of research for several decades, and it would be valuable to connect the present findings in humans, at least to some extent, with this body of work. The authors could consider discussing how the seven subfields correspond to the macaque's rostroventral (F5), caudoventral (F4), rostradorsal (F7), and caudadorsal (F2) fields as described by Geyer et al. (2000). Furthermore, the authors may reflect on the additional subdivisions of F5 (and possibly other areas) that have emerged in more recent macaque research.

Thank you for raising this important question, which relates to the homologies between human and macaque premotor areas. We agree that establishing such correspondences is crucial, and we propose that the new premotor map of the human brain presented in our study provides an improved anatomical basis for human-macaque comparisons.

A preliminary review of the literature suggests several plausible correspondences based on anatomical localization and functional properties. In the macaque, areas F7 and F2 may correspond to human areas 6d2 (pre-PMd) and 6d1 (PMd), respectively. Additionally, human area 6d3, located within the superior frontal sulcus (SFS), may correspond to macaque area F7s, which is situated within the superior arcuate branch and was mapped in the cyto- and receptorarchitectonic study by Rapan et al. (2021, NeuroImage). This correspondence is further supported by Rizzolatti et al. (1998, Electroencephalography and Clinical Neurophysiology), who identified the SFS in humans as directly comparable to the superior arcuate branch in macaques.

In macaques, the frontal eye field (FEF) is situated rostrally to the premotor cortex. Our findings suggest that human areas 6v1 and 6v2 may serve as the microstructural correlates of the FEF and the inferior FEF (iFEF), respectively. However, there remains ongoing debate about whether the human FEF or iFEF corresponds more directly to the macaque FEF (Amiez et al., 2009, Progress in Neurobiology).

Regarding ventral premotor areas, area F5 in macaques has been functionally associated with grasping actions (Rochat et al., 2010, Experimental Brain Research). This functionally aligns with human area 6v3, which we also found to be involved in grasping. There is additional evidence suggesting that human areas 6r1 and 6v3 are the putative homologues of macaque F5a and F5c (Neubert et al., 2014, Neuron; Ferri et al., 2015, NeuroImage). Notably, area F5a has been linked to language-related functions (Gerbella et al., 2011, Brain Structure and Function), which is consistent with the dysgranular structure of human area 6r1 observed in our study. Cytoarchitectonically, this positions 6r1 in close proximity to area 44, a core component of Broca's region involved in speech. Area F5p in macaques has been implicated in the hand motor network (Sharma et al., 2019, NeuroImage), further supporting its potential homology with human area 6v3. The homology of macaque area F4 remains less clearly defined. F4 is generally associated with proximal motor functions and reaching (Gentilucci et al., 1989, Brain, Behavior and Evolution). The coordinate for the human homologue of F4 suggested by Rizzolatti et al. (2002, Cognitive Neuroscience) falls within the area we define as 6v3.

While our findings offer several plausible correspondences, establishing precise homologies between human and macaque premotor areas requires a more extensive comparative analysis that integrates microstructural, functional, and connective data. Such an endeavour goes beyond the scope of the present study, but our current map provides a solid foundation for future cross-species investigations.

11. As a personal sidenote, twenty years ago in my habilitation, I summarized the discussion on the ventral/dorsal boundary of the human lateral premotor cortex as follows: “Homologies between human and monkey PMv and PMd are especially difficult to determine. This is partly due to the fact that in the monkey, cytoarchitectonic and microstimulation studies have provided conflicting evidence on the ventral-dorsal subdivision of the premotor cortex. The PMd-PMv boundary has been attributed either to the spur of the arcuate sulcus of the macaque (Rizzolatti et al., 1998, 2002) or to the inferior arcuate dimple in the owl monkey, possibly corresponding to the inferior precentral dimple of the macaque (Preuss et al., 1996). [...] According to Preuss, the human homologue of caudal PMd occupies the precentral gyrus (corresponding to area 6a α), whereas that of rostral PMd and area 8b occupies the dorsal frontal cortex rostral to the precentral sulcus (corresponding to area 6a β). Since both monkey PMv and human BA 44 are dysgranular (Bucy, 1944), and both monkey PMv as well as human BA 44 and/or ventral BA 6 represent upper limb and orofacial movements, Preuss proposes the human homologue of PMv to correspond to area 44 and the ventral part of area 6. In contrast, for Rizzolatti and co-workers (Rizzolatti et al., 1998, 2002) one important consideration is that during ontogenesis, human superior and inferior precentral sulcus develop from two separate primordia as vertical branches of the superior and inferior frontal sulcus (Turner, 1948). In view of functional differences between ventral and dorsal premotor cortex, it would be plausible to suggest this dual origin to be reflected in parallel functional differences. A crucial assumption here is that the functional areas delimited by the most ancient sulci maintain their basic location in phylogeny. This would imply that the human homologue of the superior arcuate sulcus is the superior precentral sulcus plus superior frontal sulcus. Then, dorsal area 6a α and 6a β would correspond to F2 and F7, respectively (cf. Zilles et al., 1995). The human homologue of the inferior arcuate sulcus would be the ascending branch of inferior precentral sulcus plus inferior frontal sulcus. Finally, the descending branch of inferior precentral sulcus in humans would be equivalent to the inferior precentral dimple in the monkey, and hence, human ventral area 6a α and area 44 would correspond to F4 and F5 respectively. Rizzolatti and co-workers therewith propose that human PMd is located superiorly and PMv inferiorly to about z=51 of Talairach space.” So, in the meantime, many more studies have been conducted on this topic, and while not all details may still be fully endorsed, it is nevertheless remarkable that the authors' study places the boundary between the ventral and dorsal lateral premotor cortex at the level of the extension of the SFS (superior frontal sulcus), in accordance with this argumentation. This is a very nice convergence!

Thank you for this comment. The description you provided aligns well with the findings of our study, and our interpretations and arguments support a similar direction.

Minor points:

Figure 1, part A, right hand side: “76” seems to be cut off
Addressed, new figure included.

Typos: sometimes “figure X” instead of „Figure X“

Addressed in the text.

l. 57: „Fan et al. [13]“ should read „Fan et al. [71]“ I think

Addressed. References updated.

Sex of donors of the brains was not mentioned

Information is now added in Table 4 in the Methods section.

It would be helpful to explain the abbreviations used, such as 'd' for dorsal, 'l' for lateral, 'c' for caudal, 'r' for rostral etc. (e.g., in 6dl, 6cdl, 6r). Providing this explanation once at the beginning would be sufficient.

Addressed in the Introduction (see Tab. 1).

l. 214-215 “The ventral border was identified dorsally to the LF”. Please add the ventrally bordering of area Op6 (frontal operculum), which is important to avoid a direct border with the lateral fissure. This is a particularly relevant clarification for those who don’t know that in the human precentral gyrus, the most ventral portion is not corresponding to BA 6 but to the frontal operculum.

Included in the Results section (lines 247 and 248).

Figure Legend 2: „lateral fissure preCS“ should read „lateral fissure; preCS“

Addressed in the figure legend.

l. 319-320: Regarding this point, it may be worth reconsidering the label 'attention' for the serial prediction tasks reported by Schubotz. While attention can also elicit activity in premotor areas, the premotor activations in these studies were found specifically when contrasting predictive tasks on stimulus sequences with equally attentionally demanding but non-predictive tasks (e.g., serial match-to-sample). Since attention and working memory were controlled for, they are unlikely to fully explain the observed effects. Instead, predictive task instructions on stimulus sequences or dynamic stimuli appear to be the primary drivers of premotor activity, as demonstrated across a series of studies. The authors may find a useful summary in Schubotz, 2007, Trends in Cognitive Sciences.

We have revised the Results section, Figure 6 and the Discussion accordingly, replacing “attention” by “prediction”.

Regarding the discussion of the FEF, authors may take into account the work of Tehovnik et al., 2000 (Tehovnik, E., Sommer, M., Chou, I., Slocum, W. & Schiller, P. (2000). Eye fields in

the frontal lobes of primates. *Brain Res Brain Res Rev*, 32(2-3), 413–48) and Preuss et al 1996 (Preuss, T., Stepniewska, I. & Kaas, J. (1996). Movement representation in the dorsal and ventral premotor areas of owl monkeys: a microstimulation study. *J Comp Neurol*, 371(4), 649–76).

Thanks for the references, which align well with our findings and argumentation. We have now included Tehovnik et al., 2000 and Preuss et al., 1996 in the discussion part (lines 596-598).

I. 401-402 state: “The interindividual variability in sulcal pattern, which is particularly high for the region of the junction of the preCS and the SFS.” Could you clarify what this variability is being compared to? In other words, what criterion is used to classify an anatomical landmark as exhibiting "low" versus "high" variability? Providing this context would help in better understanding the statement.

The terms low and high variability in relation to sulcal patterns (e.g., of the pMFS) are used descriptively in our study. It is well established that the frontal lobe exhibits high interindividual sulcal variability in terms of number, location, and morphological features such as continuity, interruptions, and connections with neighboring sulci (Juch et al., 2005, NeuroImage). In that study, variability was quantified using the standard deviation of sulcal coordinates along each axis, with the highest variability observed at the SFS–preCS landmark. Similarly, Drudik and Petrides (2024, Human Brain Mapping) reported substantial variability in the SFS.

In our study, interindividual variability of cortical areas is quantified using the probabilistic maps (pmaps), which reflect the spatial consistency/variability of a given area across subjects. Comparisons of variability (e.g., "lower" or "higher") are made between the areas mapped in this study. For instance, the probability maps of areas 6v1 and 6v2 contain more voxels with lower probability values (e.g., “more blue voxels”), indicating a higher degree of spatial variability compared to other areas.

To clarify this point, we have added the following sentence to the Methods section: “Descriptions such as high and low variability refer to comparisons between the areas mapped in this study, based on their respective probabilistic maps.” (lines 821-824).

I. 435-437: „These results align with monkey studies, which show overlapping representations of the hindlimb and arm in the PMd, and of the forelimb and face representation within the PMv [92]. Human functional studies have also provided evidence of a similar organisation within the human PM (e.g. [2, 93-95]).“ Authors may also consider (and possibly cite) Sakreida, K., Schubotz, R.I., Wolfensteller, U., von Cramon, D.Y. (2005). Motion class dependency in observers' motor areas revealed by functional MRI. *The Journal of Neuroscience*, 25(6), 1335-42.

Addressed. Citation included.